# cAMP signaling regulates DNA hydroxymethylation by augmenting the intracellular labile ferrous iron pool

Vladimir Camarena[1,2], David W Sant[1,2], Tyler C Huff[1,2], Sushmita Mustafi[1,2], Ryan K Muir[3], Allegra T Aron[4], Christopher J Chang[4,5,6], Adam R Renslo[3], Paula V Monje[7,8], Gaofeng Wang[1,2,9,10]*

[1]John P. Hussman Institute for Human Genomics, University of Miami Miller School of Medicine, Miami, United States; [2]Dr. John T. Macdonald Foundation Department of Human Genetics, University of Miami Miller School of Medicine, Miami, United States; [3]Department of Pharmaceutical Chemistry, University of California, San Francisco, San Francisco, United States; [4]Department of Chemistry, University of California, Berkeley, Berkeley, United States; [5]Department of Molecular and Cell Biology, University of California, Berkeley, Berkeley, United States; [6]Howard Hughes Medical Institute, University of California, Berkeley, Berkeley, United States; [7]The Miami Project to Cure Paralysis, University of Miami Miller School of Medicine, Miami, United States; [8]Department of Neurological Surgery, University of Miami Miller School of Medicine, Miami, United States; [9]Dr. Nasser Ibrahim Al-Rashid Orbital Vision Research Center, Bascom Palmer Eye Institute, University of Miami Miller School of Medicine, Miami, United States; [10]Sylvester Comprehensive Cancer Center, University of Miami Miller School of Medicine, Miami, United States

*For correspondence: gwang@med.miami.edu

Competing interests: The authors declare that no competing interests exist.

**Abstract** It is widely accepted that cAMP regulates gene transcription principally by activating the protein kinase A (PKA)-targeted transcription factors. Here, we show that cAMP enhances the generation of 5-hydroxymethylcytosine (5hmC) in multiple cell types. 5hmC is converted from 5-methylcytosine (5mC) by Tet methylcytosine dioxygenases, for which Fe(II) is an essential cofactor. The promotion of 5hmC was mediated by a prompt increase of the intracellular labile Fe(II) pool (LIP). cAMP enhanced the acidification of endosomes for Fe(II) release to the LIP likely through RapGEF2. The effect of cAMP on Fe(II) and 5hmC was confirmed by adenylate cyclase activators, phosphodiesterase inhibitors, and most notably by stimulation of G protein-coupled receptors (GPCR). The transcriptomic changes caused by cAMP occurred in concert with 5hmC elevation in differentially transcribed genes. Collectively, these data show a previously unrecognized regulation of gene transcription by GPCR-cAMP signaling through augmentation of the intracellular labile Fe (II) pool and DNA hydroxymethylation.

DOI: https://doi.org/10.7554/eLife.29750.001

## Introduction

Cyclic AMP (cAMP), the second messenger, converts the binding of ligands with many different G-protein coupled receptors (GPCRs, also known as seven-transmembrane domain receptors) into various biological activities (*Sutherland, 1970*). Principally, cAMP exerts its function through three major targets — cAMP-dependent protein kinase A (PKA), cyclic nucleotide-gated ion channels (CNGCs), and exchange protein directly activated by cAMP (Epac). It is well known that cAMP has an impact on gene transcription, which is currently considered to be mediated by the PKA-targeted

transcription factors (*Montminy, 1997*). Three transcription factors, including cAMP response element-binding protein (CREB), cAMP response element modulator (CREM), and activating transcription factor 1 (ATF1), can be phosphorylated by PKA and subsequently bind to cAMP response elements (CRE) in gene promoters, generally to activate gene transcription (*Sands and Palmer, 2008*).

Schwann cells form the myelin sheath of axons within the peripheral nervous system. cAMP is a known instructive signal for Schwann cell differentiation into a myelinating phenotype (*Jessen et al., 1991*). cAMP directly induces cell cycle arrest along with the expression of a wide variety of myelination-associated genes such as *Egr2/Krox20*, a transcription factor that is considered a master regulator of the myelin program, and myelin protein zero, the main peripheral nerve myelin protein (*Bacallao and Monje, 2015*). However, the mechanism by which cAMP regulates transcription and promotes the differentiation of Schwann cells is not fully understood.

DNA methylation is one of the major epigenetic marks that regulates gene transcription. Active DNA demethylation is catalyzed by ten-eleven translocation (Tet) methylcytosine dioxygenases, which oxidize 5-methylcytosine (5mC) to 5-hydroxymethylcytosine (5hmC) (*Tahiliani et al., 2009*; *Kriaucionis and Heintz, 2009*), and further to 5-formylcytosine (5fC) and 5-carboxylcytosine (5caC) (*He et al., 2011*; *Ito et al., 2011*). 5fC and 5caC are ultimately replaced by unmodified cytosine (5C) to complete cytosine demethylation (*Pastor et al., 2013*). It is noteworthy that 5hmC is also a unique epigenetic mark with regulatory capacities in addition to being a DNA demethylation intermediate (*Shen et al., 2014*). Tet belongs to the iron and 2-oxoglutarate (2OG, alternatively known as α-ketoglutarate)-dependent dioxygenase family which utilizes labile Fe(II) as a cofactor and 2OG as a co-substrate. We and others have shown that ascorbate, which has the capacity to reduce the redox-inactive Fe(III)/Fe(IV) to Fe(II), is another cofactor for Tet (*Minor et al., 2013*; *Yin et al., 2013*; *Blaschke et al., 2013*; *Dickson et al., 2013*; *Chen et al., 2013*). Thus, ascorbate has an impact on DNA demethylation by promoting the availability of redox-active Fe(II) to Tet. 5mC and 5hmC are major epigenetic marks that govern both cell identity and cellular phenotype transformation. Thus, it is plausible that such marks are involved in the transition of Schwann cells from the immature to myelinating phenotype.

Similar to cAMP, ascorbate was identified as another essential factor for Schwann cells to initiate and promote myelin formation (*Bunge et al., 1986*). Intracellular ascorbate deficiency can cause hypomyelination in modeled rodents (*Gess et al., 2011*). It is plausible that, by regulating DNA demethylation, ascorbate alters the cellular phenotype of Schwann cells towards a myelinating state. Based on this shared function in enhancing Schwann cell myelination, we reasoned that cAMP, like ascorbate, might also play a role in Tet-mediated DNA demethylation. Here, we investigated the potential role of cAMP in DNA demethylation. We identified that cAMP enhanced the generation of 5hmC in cultured Schwann cells. The effect of cAMP on 5hmC was not limited to Schwann cells but is likely a general effect, as it has been verified in other cell types. cAMP promptly increased the intracellular labile Fe(II) pool (LIP) independently of major cAMP targets but likely through RapGEF2 to enhance the acidification of endosomes and subsequently elevate the LIP, suggesting that an increase in labile Fe(II) over basal levels could mediate the effect of cAMP on 5hmC by activating Tet activity. The effect of cAMP on Fe(II) and 5hmC was mimicked by the addition of AC activators and phosphodiesterase (PDE) inhibitors such as caffeine. Stimulation of Schwann cells with isoproterenol or calcitonin gene-related peptide (CGRP), two ligands known to increase intracellular cAMP through Gs-coupled receptors (*Cheng et al., 1995*), elevated levels of intracellular labile Fe(II) and 5hmC. Furthermore, stimulation with cAMP caused a genome-wide shift in 5hmC profile, which correlated with a majority of the differentially expressed transcripts in Schwann cells.

## Results

### cAMP promotes 5hmc generation

In the absence of cAMP treatment, 5hmC was barely detectable by dot blot in primary cultured rat Schwann cells. However, after treatment with membrane-permeable 8-CPT-cAMP (hereafter denoted as cAMP) (100 μM), 5hmC signal emerged at comparable levels to cells treated with ascorbate (50 μM, *Figure 1A and B*). Encouraged by this initial observation, we investigated the role of cAMP on 5hmC by immunofluorescence (IF). The addition of cAMP (0–250 μM) dose-dependently

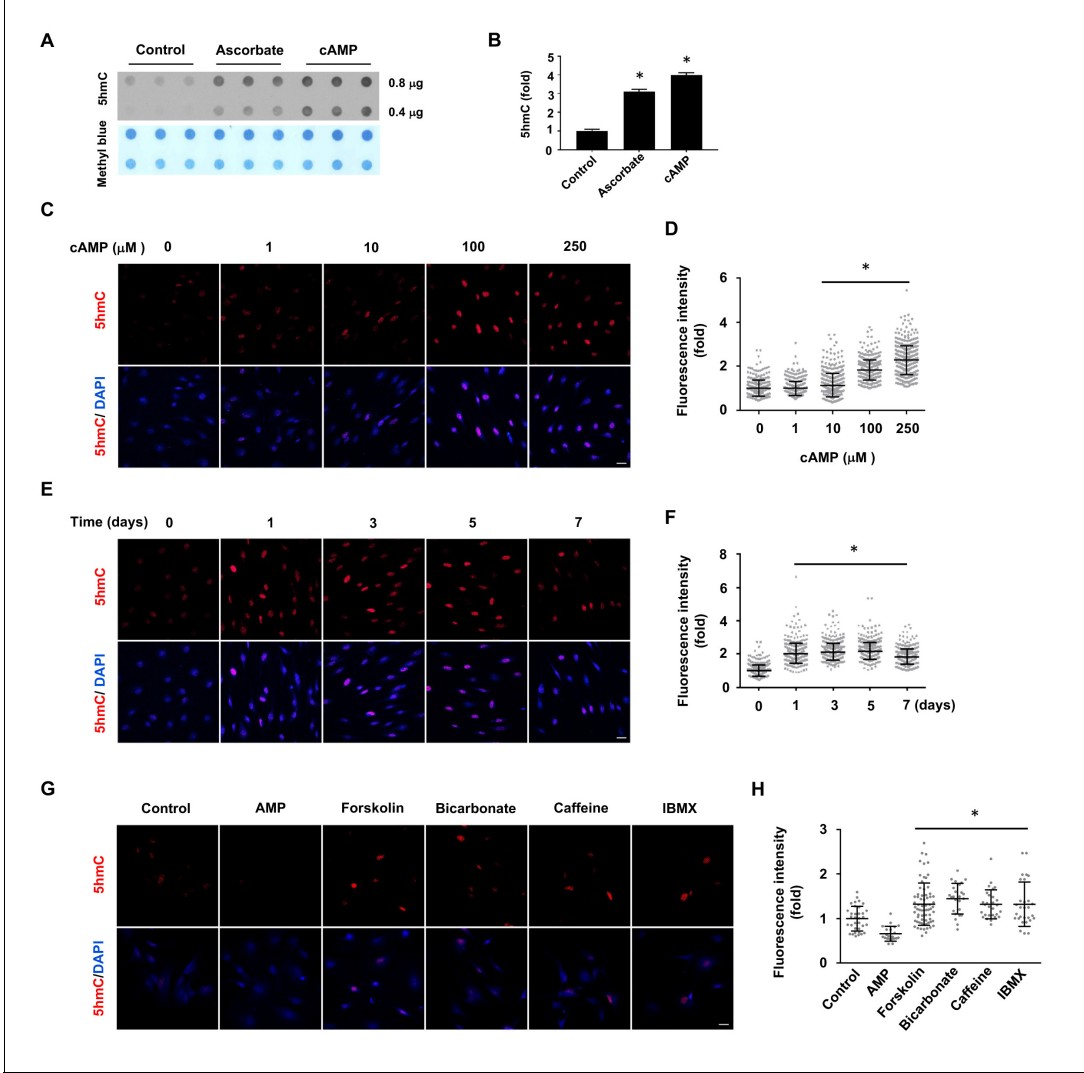

**Figure 1.** cAMP induces 5hmC in cells. (A) Dot-blot shows that treatment with ascorbate (50 µM) for 3 days or cAMP (100 µM) for 7 days induced 5hmC in Schwann cells. (B) Semi-quantification of the dot-blot shows that both cAMP and ascorbate enhanced 5hmC generation in Schwann cells. (C) cAMP (10–250 µM) increased 5hmC in Schwann cells after 7 days treatment shown by IF. (D) IF quantification indicates the dose-dependent effect of cAMP on 5hmC in Schwann cells. (E) cAMP (100 µM) treatment for 1–7 days increased 5hmC in Schwann cells. (F) IF quantification shows that the effects of cAMP treatment from 1 to 7 days induces comparable effects on 5hmC in Schwann cells. (G) Endogenous cAMP promotes 5hmC in Schwann cells. AMP (100 µM) treatment for 2 days did not induce 5hmC while AC activators (forskolin (100 µM), bicarbonate (50 mM)), and PDE inhibitors (caffeine (100 µM), IBMX (100 µM)) enhanced 5hmC generation. (H) IF quantification indicates that forskolin, bicarbonate, caffeine and IBMX increase 5hmC in Schwann cells. *p<0.0005. Scale bar = 20 µm (n = 3 independent experiments with three biological replicates in each group, error bars denote standard deviation).

DOI: https://doi.org/10.7554/eLife.29750.002

The following source data and figure supplements are available for figure 1:

**Source data 1.** Primers used for quantitative RT-PCR.
DOI: https://doi.org/10.7554/eLife.29750.007

**Figure supplement 1.** cAMP induces 5hmC in different cell types.
DOI: https://doi.org/10.7554/eLife.29750.003

**Figure supplement 2.** 5hmC elevation by shorter treatment of forskolin in Schwann cells.
DOI: https://doi.org/10.7554/eLife.29750.004

**Figure supplement 3.** 5hmC elevation by shorter treatment of cAMP and forskolin in HEK-293 cells.
DOI: https://doi.org/10.7554/eLife.29750.005

**Figure supplement 4.** Treatment with cAMP does not increase Tet transcripts.
DOI: https://doi.org/10.7554/eLife.29750.006

increased 5hmC generation in Schwann cells (*Figure 1C and D*). The IF signal for 5hmC was sustained for 1–7 days upon continuous cAMP (100 µM) treatment (*Figure 1E and F*), suggesting that continuous cAMP treatment largely maintained 5hmC. An enhanced generation of 5hmC was also verified in other cell types examined, such as HEK-293 cells, mouse embryonic fibroblasts (MEF) and neuroblastoma SH-SY5Y cells, indicating that the effect of cAMP on 5hmC is not limited to Schwann cells and is likely to be a general effect (*Figure 1—figure supplement 1*).

We next tested whether the elevation of endogenous cAMP imparts a similar effect as exogenously applied cAMP. Treating cells with forskolin, which directly activates transmembrane adenylate cyclase (AC) to produce endogenous cAMP, promoted 5hmC generation in Schwann cells as did bicarbonate, an activator of soluble AC (*Figure 1G and H*). The production of 5hmC was also observed when cells were treated with phosphodiesterase (PDE) inhibitors caffeine or IBMX, both of which prevent cAMP degradation. Conversely, no 5hmC signal was observed after cells were treated with AMP (100 µM). Collectively, these observations suggest that endogenous cAMP is indeed involved in 5hmC generation.

The increase in 5hmC generation by cAMP treatment appeared to last for days. To test whether the long-term effect on 5hmC relies on the continuous presence of cAMP or forskolin in the media, we treated Schwann cells with forskolin (10 µM) for 3–24 hr followed by washout. An increase of 5hmC was detected at both 24 or 72 hr time points, which is comparable to continuous treatment for 24 or 72 hr (*Figure 1—figure supplement 2*). Shorter treatments (1–4 hr) with cAMP (10 µM) or forskolin (10 µM) followed by washout also induced 5hmC elevation at levels comparable to continuous treatment for 24 hr in HEK-293 cells (*Figure 1—figure supplement 3*). However, unlike in Schwann cells, 5hmC level appeared to retreat toward the base line at the 72 hr time point in the fast replicating HEK-293 cells. Since 5hmC is not maintained during DNA synthesis, it is thus reasonable that 5hmC could be kept longer in the slowly dividing Schwann cells after termination of cAMP signaling. These experiments suggest that cAMP can produce a persistent increase in 5hmC, which can be detected within a few hours after treatment and last for several days depending on cell types.

## cAMP increases the intracellular labile Fe(II) pool to generate 5hmC

To understand how cAMP enhances 5hmC generation, we first examined the transcription of *Tet*. mRNA levels of *Tet2* and *Tet3* were decreased, whereas *Tet1* mRNA remained unchanged after treatment with cAMP (100 µM) for 1 day (*Figure 1—figure supplements 4* and *Figure 1—source data 1*), a time point at which cAMP clearly promoted 5hmC generation (*Figure 1E and F*). Thus, the increased level of 5hmC does not appear to be mediated by an effect of cAMP on the expression of *Tet*. In healthy cells such as primary cultured Schwann cells, the most plausible rate-limiting factor for Tet's enzymatic activity is the replenishment of the Fe(II) cofactor rather than 2OG, a relatively abundant intermediate of the Krebs cycle. We thus hypothesized that cAMP may increase access to bioavailable Fe(II), as does ascorbate. To test this hypothesis, we used two different chemical probes for assaying bioavailable Fe(II). The first probe is Trx-Puro (*Spangler et al., 2016*), a recently developed reactivity-based probe that is highly specific for redox–active labile Fe(II) within the cell. Reaction of Trx-Puro with labile Fe(II) unmasks the aminonucleoside puromycin through a Fenton-type reaction. Released puromycin is incorporated in cellular proteomes, which can be visualized using a convenient IF assay (*Spangler et al., 2016*). Treatment with cAMP at a dose as low as 1 µM increased intracellular labile Fe(II), which reached even higher levels after treatment with 10–100 µM cAMP (*Figure 2A and B*). Incubation of the cells with Diox-Puro (*Spangler et al., 2016*), a dioxolane control conjugate that is non-peroxidic, showed no IF signal after cAMP treatment indicating the Fe(II) dependence of the signal detected by Trx-Puro (*Figure 2—figure supplement 1*). The action of cAMP on intracellular labile Fe(II) was rapid. Fe(II) signal appeared within 2 hr (the minimal time for simultaneous incubation of cAMP and Trx-Puro) and peaked at 4 hr after cAMP treatment (*Figure 2C and D*). Similar to 5hmC, the effect of cAMP on intracellular labile Fe(II) was also validated in other types of cells including HEK-293, MEF, and SH-SY5Y (*Figure 2—figure supplement 2*), suggesting that it is likely a general effect.

We then applied another method to verify the effect of cAMP on labile Fe(II). The recently developed FIP-1 probe links two fluorophores through an Fe(II)-cleavable endoperoxide bridge, where Fe(II)-triggered peroxide cleavage leads to a decrease in fluorescence resonance energy transfer (FRET) from the fluorescein donor to Cy3 acceptor by splitting these two dyes into separate

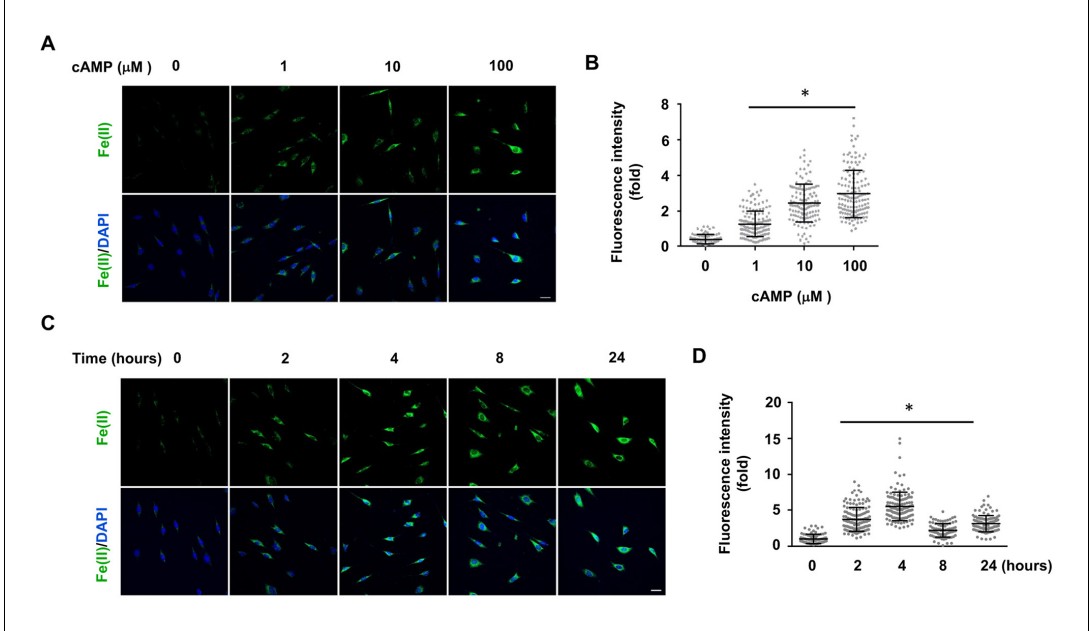

**Figure 2.** cAMP increases the intracellular labile Fe(II) pool in cells. (A) cAMP (1–100 μM) treatment for 4 hr increased the intracellular labile Fe(II) pool detected by Trx-Puro ferrous iron probes. (B) IF quantification shows the dose-dependent effect of cAMP on labile Fe(II). (C) cAMP (100 μM) treatment for 2–24 hr increased labile Fe(II). (D) IF quantification shows the peak effect of cAMP on labile Fe(II) after treatment for 4 hr. Scale bar = 20 μm. *p<0.0005 (n = 3 independent experiments with three biological replicates in each group, error bars denote standard deviation).

DOI: https://doi.org/10.7554/eLife.29750.008

The following figure supplements are available for figure 2:

**Figure supplement 1.** Negative control for puromycin incorporation with labile Fe(II) probe TRX-puro in Schwann cells.

DOI: https://doi.org/10.7554/eLife.29750.009

**Figure supplement 2.** cAMP increases the intracellular labile Fe(II) pool in different cell types.

DOI: https://doi.org/10.7554/eLife.29750.010

**Figure supplement 3.** (A) Representative ratiometric confocal microscopy images of live HEK-293 cells loaded with FIP-1.

DOI: https://doi.org/10.7554/eLife.29750.011

fragments (*Aron et al., 2016*). Using the FIP-1 probe, a significant increase of labile Fe(II) was detected in cells after treatments with cAMP (10, 100 μM) for 4 hr (*Figure 2—figure supplement 3*). Taken together, the results from two different independent chemical methods verified that cAMP treatment likely elevated labile Fe(II) in the cell.

Treatment with AMP had no observable effect on labile Fe(II). Elevating endogenous cAMP also augmented intracellular labile Fe(II), as demonstrated by the treatment of Schwann cells with AC activators (forskolin, bicarbonate) and PDE inhibitors (caffeine, IBMX) (*Figure 3A and B*). To further examine the time course of labile iron response to cAMP, we treated Schwann cells with forskolin (10 μM) for 3 hr followed by washout. Labile Fe(II) immediately increased after the treatment but quickly declined back to the baseline levels after another 3 hr (*Figure 3—figure supplement 1*). These results suggest that the effect of cAMP on labile Fe(II) elevation is relatively transient. Alternatively, intracellular labile Fe(II) might be tightly controlled within the cell and therefore any increase in the LIP would not last a long time.

Ascorbate at physiological levels has been shown to have no obvious influence on intracellular labile Fe(II) (*Spangler et al., 2016*). Indeed, no obvious change in the intracellular labile Fe(II) pool was observed in Schwann cells after treatment with ascorbate at a physiological concentration (50 μM) (*Figure 3—figure supplement 2*), suggesting that ascorbate may only increase the bioavailability of Fe(II) to Tet and other enzymes locally.

To determine whether cAMP promotes 5hmC generation via labile Fe(II), Schwann cells were treated with two different iron chelators 2,2, bipyridyl (20 μM) and deferoxamine (20 μM) prior to cAMP stimulation. These iron chelators drastically decreased the level of labile Fe(II) and further

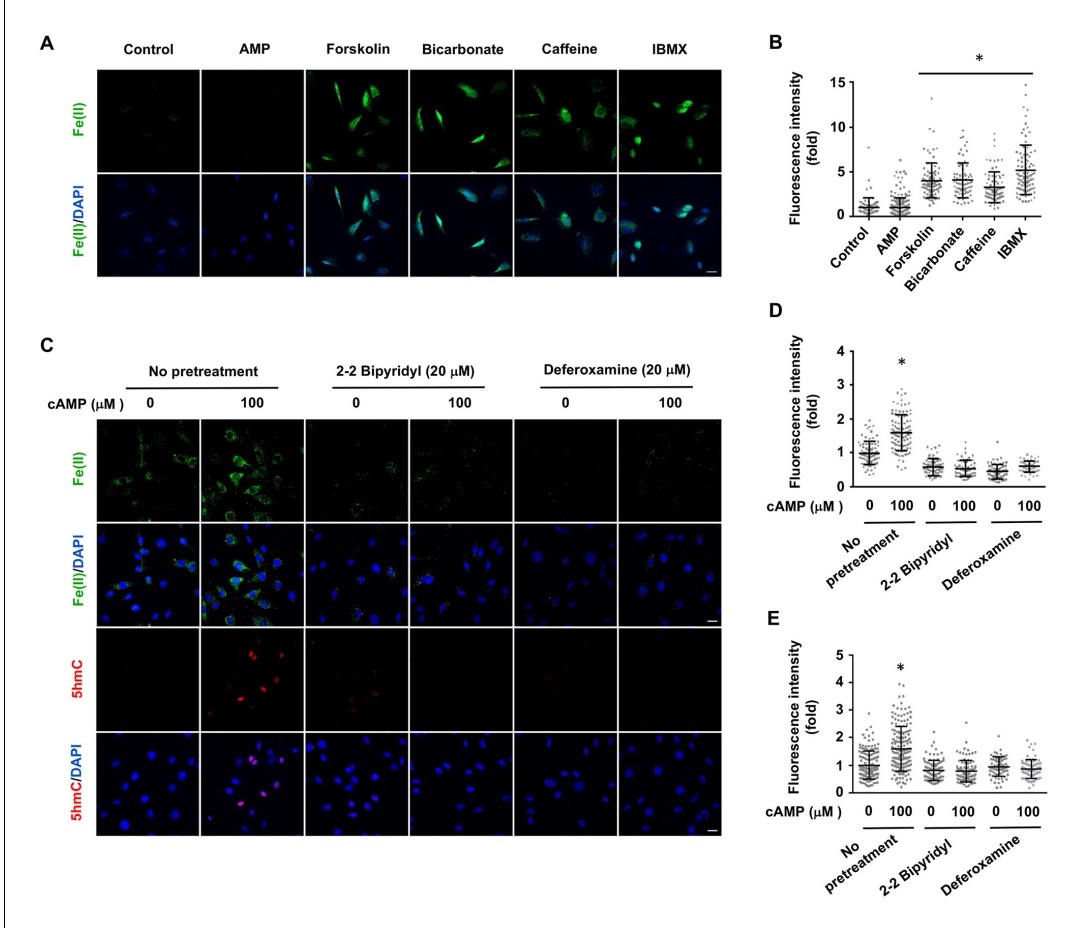

**Figure 3.** Induction of labile Fe(II) by endogenous cAMP and the dependency of 5hmC generation on labile Fe(II). (**A**) AMP (100 µM) treatment for 4 hr did not induce labile Fe(II) while AC activators (forskolin (100 µM), bicarbonate (50 mM)) and PDE inhibitors (caffeine (100 µM), IBMX (100 µM)) increased labile Fe(II) detected by Trx-Puro probes. (**B**) IF quantification indicates the elevated labile Fe(II) by forskolin, bicarbonate, caffeine, and IBMX. (**C**) Pretreatment with iron chelators 2,2, bipyridyl (20 µM) and deferoxamine (20 µM) for 20 min blocked the elevation of labile Fe(II) and 5hmC in Schwann cells by cAMP (100 µM) treatment. (**D**) IF quantification indicates the inhibition of iron chelators on the induction of labile Fe(II) by cAMP. (**E**) IF quantification indicates the inhibition of iron chelators on the induction of 5hmC by cAMP. Scale bar = 20 µm. *p<0.0005 (n = 3 independent experiments with three biological replicates in each group).

DOI: https://doi.org/10.7554/eLife.29750.012

The following figure supplements are available for figure 3:

**Figure supplement 1.** Relatively transient increase intracellular labile Fe(II) by short forskolin treatments.
DOI: https://doi.org/10.7554/eLife.29750.013

**Figure supplement 2.** Ascorbate does not increase the intracellular labile Fe(II) pool in Schwann cells detected by Trx-Puro probes.
DOI: https://doi.org/10.7554/eLife.29750.014

**Figure supplement 3.** Pretreatment with iron chelators 2,2, bipyridyl (20 µM) and deferoxamine (20 µM) for 20 min blocked the upregulation of 5hmC by cAMP (100 µM) treatment in A2058 cells.
DOI: https://doi.org/10.7554/eLife.29750.015

abolished the effect of cAMP on 5hmC in Schwann cells (*Figure 3C and D*). The abolishing effect of the iron chelators on cAMP-induced 5hmC generation was also verified in other cells such as human melanoma A2058 cells (*Figure 3—figure supplement 3*). These results suggest that cAMP upregulates 5hmC by elevating intracellular labile Fe(II).

## cAMP increases the intracellular labile Fe(II) by enhancing endosome acidification.

The molecular mechanism by which cAMP alters labile iron in principle could be related to iron uptake and storage in the cell. Cellular uptake of iron is a complicated cascade involving the internalization of transferrin-transferrin receptor complex and Fe(III) discharge from transferrin after the acidification of endosomes, via vacuolar $H^+$-ATPase (V-ATPase, the $H^+$ pump). Subsequently, Fe(III) is converted to Fe(II) by Steap3 and leaves the endosome via divalent metal transporter 1 (DMT1) to enter the LIP (*De Domenico et al., 2008*). One key step in iron uptake is endosome acidification. cAMP treatment consistently decreased the pH in endocytotic vesicles as measured using vesicle pH indicator that emits increasing levels of fluorescence as pH decreases from 8 to 4 (*Figure 4A*). These results suggest that cAMP enhances the acidification of endosomes, which could underlie the elevated LIP.

We then tested whether elevated labile Fe(II) and 5hmC by cAMP signaling is mediated by enhanced endosome acidification. Cells were pretreated with V-ATPase inhibitor Bafilomycin A1 (200 nM). Indeed, Bafilomycin A1 largely abolished the effect of cAMP on labile Fe(II) as well as

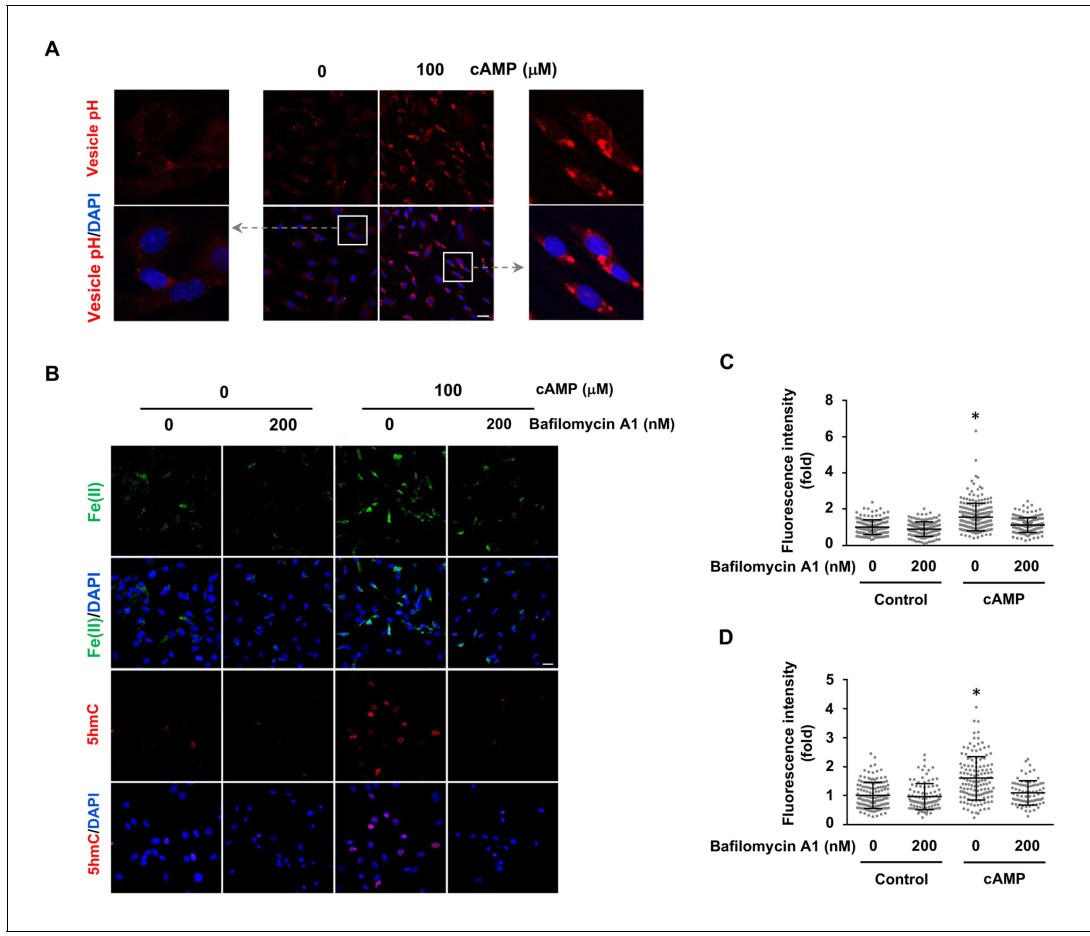

**Figure 4.** cAMP increases labile Fe(II) and 5hmC by enhancing the acidification of endosomes. (**A**) cAMP (100 µM) treatment decreases the pH in intracellular vesicles. Red fluorescence increases when the pH decreases from 8 to 4. (**B**) V-ATPase inhibitor Bafilomycin A1 (200 nM) pretreatment for 30 min blocks the effect of cAMP on both labile Fe(II) detected by Trx-Puro probes and 5hmC in Schwann cells. (**C**) IF quantification indicates the inhibition of Bafilomycin A1 on labile Fe(II) elevation by cAMP. (**D**) IF quantification indicates the inhibition of Bafilomycin A1 on 5hmC induction by cAMP. Scale bar = 20 µm. p<0.0005 (n = 3 independent experiments with three biological replicates each).

DOI: https://doi.org/10.7554/eLife.29750.016

The following figure supplement is available for figure 4:

**Figure supplement 1.** Decreasing the expression of Ferritin does not block the induction of labile Fe(II) by cAMP.

DOI: https://doi.org/10.7554/eLife.29750.017

5hmC in the cells (*Figure 4B–D*). Furthermore, knocking down the expression of ferritin did not block the induction of labile Fe(II) elevation by cAMP indicating that iron storage may not be a major target of cAMP signaling (*Figure 4—figure supplement 1*). Overall, these results suggest that cAMP signaling upregulates the LIP, and subsequently 5hmC generation by acidifying the endosome.

In an attempt to understand how cAMP signaling causes endosome acidification, we examined the major known targets of cAMP, which include PKA, CNGCs, and Epac. PKA phosphorylates substrate proteins and Epac acts as a guanine nucleotide exchange factor for the small G protein Rap. CNGCs are nonselective cation channels for $Na^+$, $Ca^{++}$, and other cations, but have no known role in transporting iron. To determine whether labile Fe(II) depends on the major known signaling targets of cAMP, we pretreated the cells with PKA inhibitors (KT5720, H89), an Epac inhibitor (ESI09) or a CNGC channel blocker (LCD). Neither the inhibitors nor the channel blocker prevented the observable alterations in labile Fe(II) in cAMP-treated Schwann cells (*Figure 5A*). Using an in vitro assay, we confirmed that H89 (20 μM) and KT5720 (2 μM) indeed abolished the cAMP-induced phosphorylation of Peptag peptide which is catalyzed by PKA, thus confirming the potency of the two PKA inhibitors (*Figure 5—figure supplement 1*). Furthermore, the transcripts of CNGC and Epac (*RapGEF3*, *RapGEF4*) were expressed at either null or extremely low levels in control and cAMP-treated Schwann cells (*Figure 5—source data 1*), suggesting that the impact of CNGC and Epac on intracellular labile Fe(II) is likely to be negligible. These results suggest that the upregulation of intracellular labile Fe(II) by cAMP is likely independent of these known pathways.

Although the transcription of Epac1 (*RapGEF3*) and Epac2 (*RapGEF4)* is very low, other *RapGEFs*, especially *RapGEF2* transcripts are expressed at a much higher level (>330 fold of Epac1 or Epac2) in Schwann cells. RapGEF2 appears to be also a target of cAMP, which could be further involved in the activation of other small GTPases (*Pak et al., 2002*). To explore whether RapGEF2 mediates the elevation of labile Fe(II) via endosome acidification, we used siRNA to knockdown its expression. Due to the low transfection efficiency of primary cultured Schwann cells, we alternatively used HEK-293 cells, which also showed a robust elevation of labile Fe(II) and 5hmC in response to cAMP treatment. In scramble siRNA treated cells, cAMP treatment decreased the pH value in endocytic vesicles and elevated the intracellular labile Fe(II). However, after the expression of RapGEF2 was reduced by siRNA, the effect of cAMP on endosome acidification and labile Fe(II) was largely abolished (*Figure 5B–D*). The knockdown effect of siRNA on RapGEF2 was confirmed by qRT-PCR (*Figure 5— figure supplement 2*). Rap proteins are the major effectors of RapGEF2. To test whether Rap is involved in labile Fe(II) induction by cAMP, we knocked down Rap isoforms by siRNA. Knocking down the expression of Rap1, but not Rap2, largely abolished the vesicle acidification of vesicles and labile Fe(II) elevation induced cAMP (*Figure 5—figure supplement 3*). These results suggest that RAP1 could be one of the downstream effectors of RapGEF2 mediating the signal that elevates intracellular labile Fe(II) in response to cAMP. Overall, these results suggest that cAMP signaling, via RapGEF2 and Rap1, acidifies the endosome to augment the LIP.

## Activation of Gs-coupled receptors increases intracellular labile fe(II) and 5hmC

Signaling of many GPCRs either up- or downregulates intracellular cAMP, depending on the coupled Gs or Gi. It was shown that isoproterenol and calcitonin gene-related peptide (CGRP) elevate cAMP in Schwann cells by binding to β-adrenergic receptors and CGRP receptors respectively (*Cheng et al., 1995*), both of which are coupled with Gs. We therefore used these two GPCRs as models to test whether their signaling can change DNA hydroxymethylation. Indeed, stimulation with isoproterenol or CGRP increased 5hmC in Schwann cells (*Figure 6A and B*), which correlated with an elevated level of intracellular labile Fe(II) (*Figure 6C and D*). Furthermore, AC inhibitor SQ22536 largely abolished the effect of isoproterenol and CGRP on labile Fe(II) (*Figure 6C and D*). Overall, these results suggest that GPCR signaling might regulate DNA hydroxymethylation by augmenting the intracellular Fe(II) pool via activated ACs and elevated cAMP.

## cAMP changes the 5hmC profile in differentially expressed genes.

It has been previously shown that cAMP shifts the transcriptome (*Montminy, 1997*). Indeed, 7020 transcripts were differentially transcribed in Schwann cells in response to elevated cAMP as shown

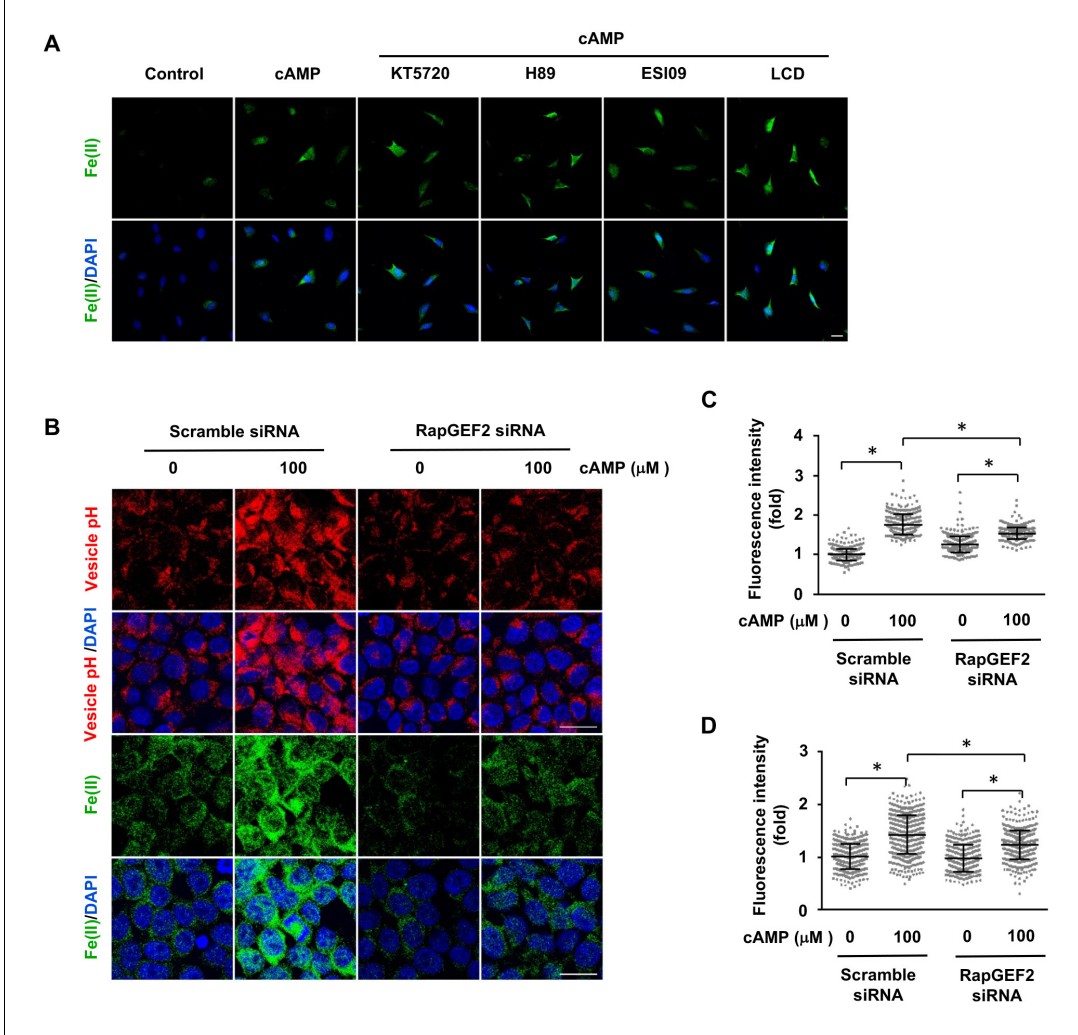

**Figure 5.** The regulation of labile Fe(II) by cAMP is likely mediated via RapGEF2. (**A**) Pretreatment with PKA inhibitors (KT5720 (2 µM), H89 (20 µM)), Epac inhibitor ESI09 (10 µM) or CNGC blocker LCD (10 µM) for 20 min prior to cAMP addition showed no effect on labile Fe(II) induced by cAMP (100 µM) treatment in Schwann cells detected by Trx-Puro probes. (**B**) Knocking down the expression of RapGEF2 in HEK-293 cells largely blocked the acidification of vesicles and labile Fe(II) elevation after cAMP (100 µM) treatment compared to the scramble siRNA group. (**C**) IF quantification indicates that knocking down RapGEF2 inhibits but does not completely abolish vesicle acidification induced by cAMP. (**D**) IF quantification indicates that knocking down RapGEF2 inhibits but does not completely abolish labile Fe(II) induced by cAMP. Scale bar = 20 µm. p<0.0005 (n = 3 independent experiments with three biological replicates in each group).

DOI: https://doi.org/10.7554/eLife.29750.018

The following source data and figure supplements are available for figure 5:

**Source data 1.** Fragments per kilobase per million (FPKM) of CNG and Rapgef genes in Schwann cells.
DOI: https://doi.org/10.7554/eLife.29750.022

**Figure supplement 1.** Abolishment of PKA activity by inhibitors H89 and KT5720.
DOI: https://doi.org/10.7554/eLife.29750.019

**Figure supplement 2.** qRT-PCR shows that the mRNA level of RapGEF2 is lower in the siRNA group compared to the scramble siRNA group (p=0.042) (n = 3 independent experiments with three biological replicates each, error bars denote standard error).
DOI: https://doi.org/10.7554/eLife.29750.020

**Figure supplement 3.** RAP1 and labile Fe(II) induction by cAMP.
DOI: https://doi.org/10.7554/eLife.29750.021

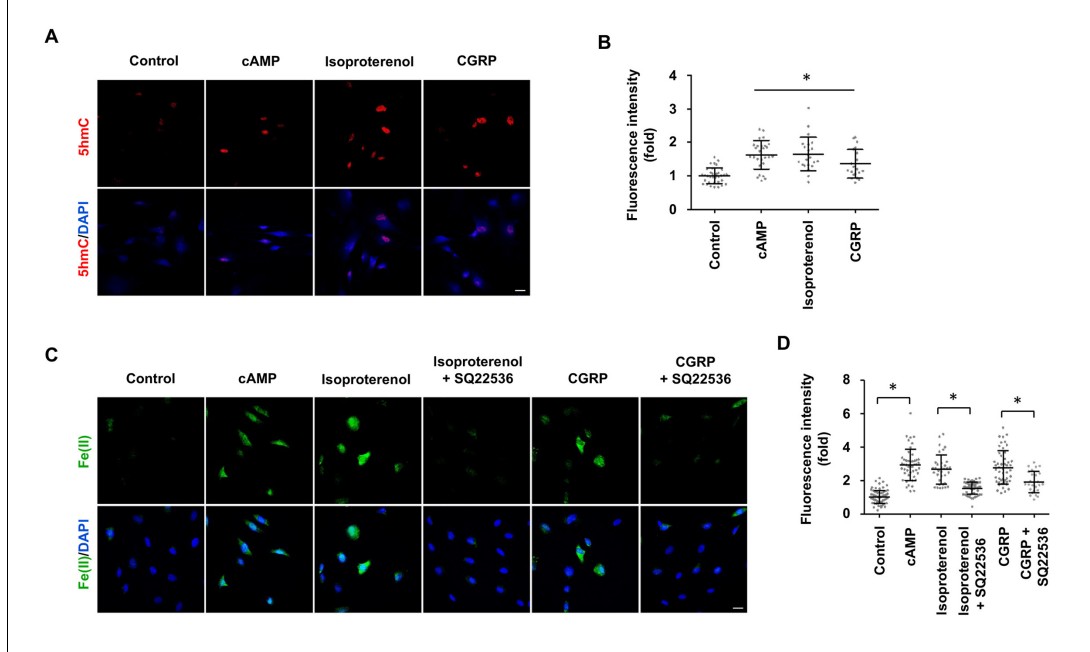

**Figure 6.** GPCR stimulation induces labile Fe(II) and 5hmC generation in Schwann cells. (**A**) Treatment with Gs-coupled receptor ligands isoproterenol (10 μM) and CGRP (100 nM) for 2 days induced 5hmC, an effect comparable to treatment with cAMP (100 μM) as shown by IF. (**B**) IF quantification indicates that activation of Gs-coupled receptors elevates 5hmC. (**C**) Treatment with isoproterenol (10 μM) and CGRP (100 nM) for 4 hr enhanced intracellular labile Fe(II) detected by Trx-Puro probes, which is comparable to treatment with cAMP (100 μM). The induction of labile Fe(II) by isoproterenol and CGRP was decreased after pretreatment with SQ22536 (100 μM) for 20 min prior to isoproterenol and CGRP stimulation. Scale bar = 20 μm. (**D**) IF quantification indicates that SQ22536 inhibits the elevation of labile Fe(II) by Gs-coupled receptor ligands. p<0.0005 (n = 3 independent experiments with three biological replicates in each group).

DOI: https://doi.org/10.7554/eLife.29750.023

by RNA-seq (*Figure 7A and B*). Of the differential transcripts, 54% were upregulated and 46% downregulated, which is concordant with the bi-directional transcriptional regulation of 5hmC (*Wu et al., 2011*). Furthermore, genome-wide 5hmC profiles were also altered by cAMP treatment as revealed by hMeDIP-seq (*Figure 7C*). In total, 66,963 5hmC peaks were upregulated and 10,026 peaks were downregulated by cAMP treatment. By integrating RNA-seq and hMeDIP-seq, we found that 4071 differential transcripts (58% of total differential transcripts) correlate with altered 5hmC peaks located within promoter regions or gene bodies (*Figure 8A*). Overall, cAMP increased 5hmC levels mainly in gene bodies globally in these differential transcripts. However, the 5hmC increase was much more dramatic in upregulated genes compared to downregulated genes. Comparatively, there was no obvious change of 5hmC in the promoters of transcriptionally unchanged genes (*Figure 8—figure supplement 1*). These results suggest that changes in 5hmC, especially in gene bodies, could be responsible for the differential transcription or involved in its regulation.

Currently, it is thought that PKA-dependent phosphorylation of three transcription factors including cAMP response element-binding protein (CREB), cAMP response element modulator (CREM), and activating transcription factor 1 (ATF1) are responsible for the transcriptional changes caused by cAMP (*Sands and Palmer, 2008*). However, 3965 transcripts (56.5% of total differential transcripts) might be targeted by the three transcription factors after reviewing ChIP data in ENCODE and ChEA databases (*Yip et al., 2012*; *Lachmann et al., 2010*). Of the 3965 transcripts, 2372 transcripts were also accompanied with differential 5hmC peaks, suggesting that these genes might be dually regulated by 5hmC and the transcription factors (*Figure 8B*). Additionally, 24.2% of differentially transcribed genes have differential 5hmC peaks but have not been previously shown to be targeted by CREB, CREM, or ATF1. The transcription of 19 genes with known functions in regulating Schwann cell myelination was affected by cAMP. Overall, 13 pro-myelinating genes were upregulated and six anti-myelinating genes were downregulated by cAMP treatment (*Figure 8—source data 1*). For example, cAMP dramatically induced the transcription of *Egr2* (also known as *Krox-20*), which plays

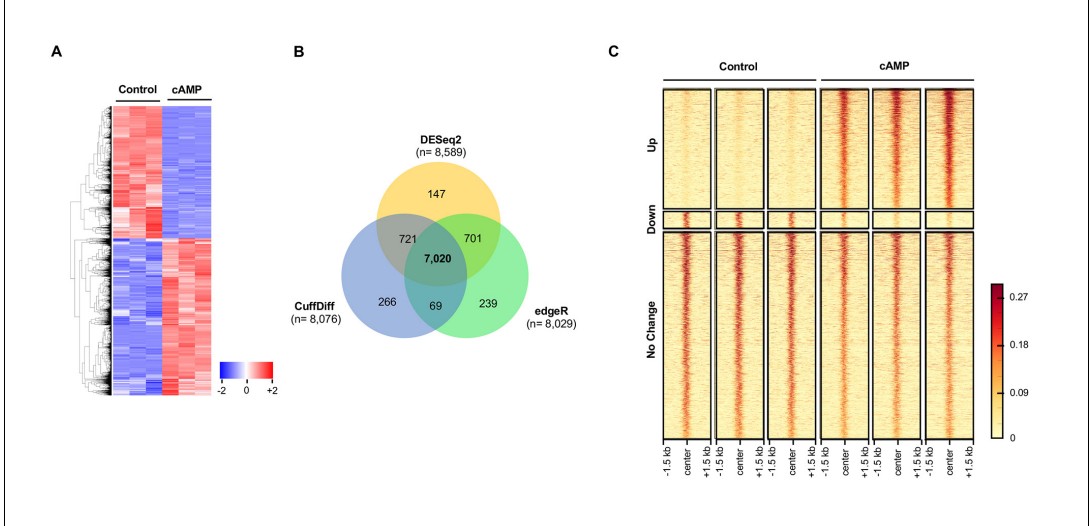

**Figure 7.** cAMP shifts the transcriptome and hydroxymethylome of Schwann cells. (**A**) cAMP (100 μM) treatment for 7 days changes genome-wide transcription as shown by the heatmap of the relative abundance of reads for differential transcripts. (**B**) Venn Diagram of the RNA-seq results from the comparison of Schwann cells treated with or without cAMP (100 μM) for 7 days. The numbers within the circles represent the number of differential transcripts called by DESeq2 (yellow), edgeR (green), and CuffDiff (Blue) respectively. (**C**) cAMP (100 μM) treatment upregulated 5hmC peaks genome-wide as detected by hMeDIP-seq (n = 1 experiment with three biological replicates).

DOI: https://doi.org/10.7554/eLife.29750.024

a key role in Schwann cell myelination (*Arthur-Farraj et al., 2011*). After cAMP treatment, there was an obvious increase of 5hmC peaks in the gene body of *Egr2* (*Figure 8C*), which is also thought to be targeted by CREB (*Hossain et al., 2012*). On the other hand *Pmp2*, which is predominantly expressed in myelinating Schwann cells (*Zenker et al., 2014*), has no obvious binding motif for the PKA-dependent transcription factors. cAMP treatment elevated the 5hmC level in the gene body and upregulated its transcription dramatically, suggesting this gene could be regulated by 5hmC and not the PKA-dependent transcription factors.

## Discussion

It is known that epigenetic changes, such as DNA methylation, reflect the interface of a dynamic extracellular microenvironment and the genome. However, the specific signaling molecules that serve as mediators between environmental variation and epigenomic changes remain largely elusive. On the other hand, GPCRs, the largest and most diverse group of membrane receptors, sense extracellular changes by binding with specific ligands. Signaling of cAMP, one major second messenger of the GPCRs, and its targets has been thoroughly studied and was thought to be well established. Our results show a previously unknown pathway of GPCR-cAMP signaling in promoting the intracellular labile Fe(II) pool, enhancing DNA hydroxymethylation and consequently changing gene transcription.

The convergence of 5hmC generation and labile Fe(II) concentrations suggests a previously unrecognized role for labile Fe(II) in propagating a signal initiated by cAMP and mediated by the Fe(II)-dependent oxidase activity of Tet. This new role for iron is a plausible one given that cellular iron uptake and the shuttling of iron from storage in ferritin as Fe(III) to its entry into the cytosolic labile pool of Fe(II) is a fundamental aspect of cellular iron metabolism. While this process is clearly involved in maintaining iron homeostasis and the biosynthesis of Fe(II) cofactors, it appears also to have been appropriated for cellular signaling in the particular case of enzymes that employ labile Fe(II) as an essential co-factor, such as the iron and 2OG-dependent Tet and Jumonji C domain-containing histone demethylases. The changes to labile cytosolic Fe(II) measured by Trx-Puro likely also reflect the nuclear Fe(II) pool, since there is a rapid equilibrium between these two Fe(II) pools,

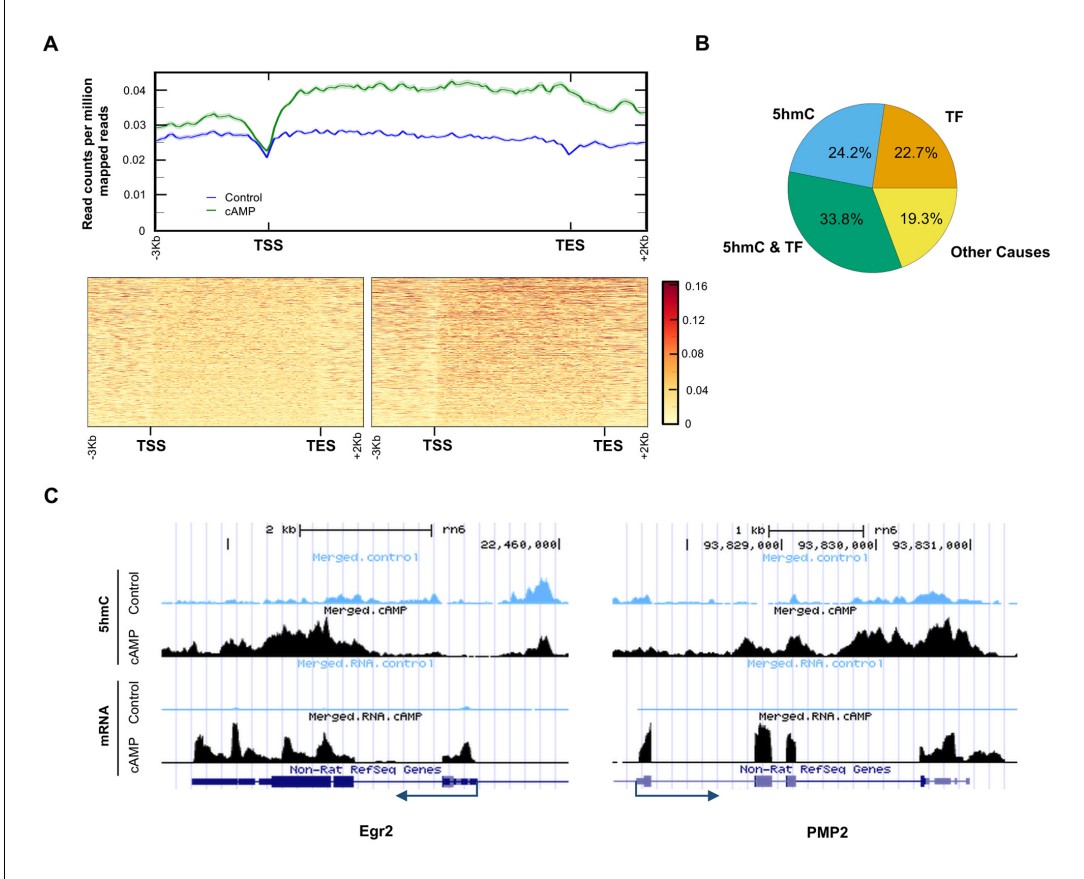

**Figure 8.** 5hmC profile changes correlate with gene transcription. (**A**) cAMP (100 μM) upregulated 5hmC at promoter and gene body regions of differential transcripts. (**B**) 33.8% of differential transcripts were associated with both 5hmC peaks and PKA-dependent transcription factors (TF). 24.2% of differential transcripts were associated with 5hmC peaks only and 22.7% of differential transcripts were associated with TF only. The rest of the differential transcripts (19.3%) were not associated with either 5hmC peaks or TF. (**C**) cAMP (100 μM) increased transcription and 5hmC, mainly in the gene bodies, of *Egr2* and *Pmp2* shown by UCSC Genome Browser views of hMeDIP-seq and RNA-seq reads (n = 1 experiment with three biological replicates).

DOI: https://doi.org/10.7554/eLife.29750.025

The following source data and figure supplement are available for figure 8:

**Source data 1.** Impact of cAMP treatment on both transcription and 5hmC levels of myelin-related genes in Schwann cells.
DOI: https://doi.org/10.7554/eLife.29750.027

**Figure supplement 1.** Analysis of hMeDIP-seq enrichment levels in the promoter region (3,000 bp upstream of TSS) and gene body regions (TSS to TES) of genes that change in response to cAMP treatment and nondifferential genes.
DOI: https://doi.org/10.7554/eLife.29750.026

presumably via nuclear pores (*Ma et al., 2015*). Thus, the results described above suggest that cAMP increases the accessibility of Fe(II) to Tet by augmenting labile Fe(II) in the cytosol and nucleus.

An increase in labile Fe(II) can be induced by cAMP in as little as 2 hr, indicating that no new protein synthesis is required. The molecular mechanism by which cAMP alters labile Fe(II) appears to be related to iron uptake rather than release from storage in ferritin. Cellular iron uptake is a multistep process including internalization of the transferrin-transferrin receptor complex, vesicle formation, endosome acidification, Fe(III) discharge, conversion of Fe(III) to Fe(II), and release of Fe(II) from the endosome to the LIP. cAMP could potentially be involved in each step of iron uptake. Our results suggest that cAMP signaling increases the acidification of endocytotic vesicles, which is essential for Fe(III) discharge from the internalized transferrin. This is consistent with earlier studies showing that cholera and pertussis toxins, by increasing intracellular cAMP, enhance endosome acidification (*Van Dyke, 1997*). It could be that cAMP signaling increased the number and/or activity of V-ATPase

to pump $H^+$ into the endosome to release Fe(II) to the LIP, which subsequently promotes 5hmC generation. Indeed, we showed that inhibitors of V-ATPase largely abolish the effect of cAMP on labile Fe(II) and 5hmC.

The elevation of labile Fe(II) by cAMP signaling appears independent of the major targets of cAMP including PKA, Epac, and CNGC. We examined the effect of RapGEF2 initially based on its higher expression level in Schwann cells. cAMP failed to induce labile Fe(II) and 5hmC once the expression of RapGEF2 or Rap1 was reduced, suggesting that RapGEF2 and its downstream effector Rap1 could be at least major mediators of cAMP to regulate labile Fe(II) and 5hmC. Thus, it is plausible that RapGEF2, directly or indirectly, affects the number and/or activity of V-ATPase in the endosome. Future studies are warranted to investigate how RapGEF2 and Rap1 regulate the function of V-ATPase.

Both cAMP and ascorbate appear to enhance the activity of Tet to generate 5hmC. However, the mechanisms of cAMP and ascorbate in promoting the availability of Fe(II) to Tet are likely different. Ascorbate at physiological concentrations has no obvious influence on intracellular labile Fe(II) shown here and before (*Spangler et al., 2016*), but is thought to maintain local iron in the active form of Fe (II) for Tet as it does for other hydroxylases (*Mandl et al., 2009*). In contrast, cAMP elevates the global intracellular labile Fe(II) pool.

For the first time, GPCRs via cAMP are directly linked to epigenetic regulation. The first hint of a possible role for cAMP in DNA hydroxymethylation came from studies on Schwann cells. Using Schwann cells as a model, we showed that intracellular cAMP elevation increased the intracellular labile Fe(II) pool, changed genome-wide 5hmC profile, and altered the transcription of many genes. More importantly, activation of different Gs-coupled receptors demonstrated increases in intracellular labile Fe(II) and 5hmC similarly to treatments that elevate intracellular cAMP including membrane permeable cAMP, AC activators, or PDE inhibitors. Coupled with Gs, Gi, or both, GPCRs can raise or decrease the intracellular cAMP, which will further change DNA methylation-demethylation dynamics. Thus GPCRs, via cAMP, transduce microenvironmental cues to epigenetic alterations.

One recent study showed that cAMP has an impact on DNA methylation by mainly increasing the expression of Tet or DNA methyltransferases (*Fang et al., 2015*). This may not be a general effect but more likely a cell-specific effect since we observed a decreased expression of Tet2 and Tet3 in Schwann cells after cAMP treatment.

In this study, most experiments were conducted in Schwann cells. However, the effect of cAMP on 5hmC is not limited to Schwann cells and is likely a general effect, as it has been verified in other cell types. Studies have shown that 5hmC is not only a demethylation intermediate but also a unique epigenetic mark that regulates transcription by recruiting a different set of binding proteins compared to 5mC (*Spruijt et al., 2013*). Indeed, altered 5hmC profiles caused by cAMP correlated with a majority of differentially transcribed genes. Thus, cAMP regulates the transcriptome by enhancing DNA demethylation, in addition to the known pathway of PKA-targeted transcription factors. In silico analysis suggests that some differential transcripts correlate with either 5hmC or the three transcription factors, while other transcripts correlate with both or none of them. The focus of this study is to connect the dots from GPCR agonists to cAMP, elevated Fe(II), 5hmC formation, and ultimately transcriptional effects. Future studies can dissect the impact of cAMP on the transcriptome and the contribution of 5hmC and/or the three transcription factors in different cell types.

In conclusion, our results show a novel function of cAMP signaling in regulating DNA hydroxymethylation by increasing the intracellular labile Fe(II) pool likely via a non-canonical pathway, which consequently promotes the oxidation of 5mC to 5hmC by Fe(II)-dependent Tet. Dynamic intracellular cAMP, regulated by Gs-/Gi-coupled receptors and factors affecting AC/PDE directly, therefore could have a profound impact on the epigenome.

## Materials and methods

### Materials

cAMP [8-(4-Chlorophenylthio) adenosine-3',5'-cyclic monophosphate] and AMP (8- Bromoadenosine-5-O-monophosphate) were purchased from Biolog (US distributor, Axxora LLC, San Diego, CA). Recombinant heregulin-β1 (herein referred to as neuregulin) was purchased from Peprotech (Rocky Hill, NJ). Forskolin, poly-l-lysine, bicarbonate, sodium ascorbate, bafilomycin A1, 2,2'-bipyridyl,

deferoxamine and laminin from Engelbreth-Holm-Swarm murine sarcoma basement membrane were purchased from Sigma-Aldrich (St. Louis, MO). Caffeine was bought from Enzo Life Sciences (Farmingdale, NY). SQ 22536, IBMX, ESI-09, CGRP (calcitonin gene-related peptide (rat)) and isoproterenol hydrochloride were purchased from Tocris Bioscience (Bristol, UK). KT5720 and H-89 Dihydrochloride were purchased from Millipore (Billerica, MA). LCD (L-cis-Diltiazem) was purchased from Abcam (Cambridge, UK). Dulbecco's modified eagles's medium (DMEM) that contain reduced level of sodium bicarbonate (1.5 g/L) (cat#30–2002) was purchased from American Type Culture Collection (ATCC) (Manassas, VA).

## Cell culture and treatments

Primary cultured Schwann cells were isolated from the sciatic nerves of 3 month-old Fisher rats as described previously (*Bacallao and Monje, 2015*). Briefly, sciatic nerves were cut into small segments and allowed to degenerate in vitro by incubation for 10 days in DMEM medium containing 10% heat inactivated FBS. Degenerated nerve explants were dissociated with a mixture of 0.25% dispase and 0.05% collagenase and the resulting cell suspension was plated on poly-L-lysine (PLL)-coated dishes. The purified Schwann cells were expanded up to passage one in DMEM supplemented with 2 µM forskolin and 10 nM neuregulin. Experiments were performed on Schwann cells between 3 to 5 rounds of expansion plated on PLL-laminin coated glasses in 24-well dishes. The media used were DMEM supplemented with 10% FBS and without forskolin and neuregulin. Treatment of Schwann cells with cAMP was carried out as reported previously (*Bacallao and Monje, 2015*). HEK-293, A2058, and SH-SY5Y cell lines were obtained from ATCC (Manassas, VA). Mouse embryonic fibroblasts were a gift from Dr. Katherine Walz from the University of Miami. All these cells were cultured in DMEM containing 10% heat-inactivated FBS. Each treatment group contains three wells and major experiments were replicated at least three times.

## 5hmc dot-blot assay

Genomic DNA was extracted from cultured Schwann cells using QIAamp DNA mini kits (Qiagen, Hilden, Germany). A Qubit Fluorometer (Life Technologies, Carlsbad, CA) was used to quantify the concentration of DNA. DNA samples were diluted with 2N NaOH and 10 mM Tris·Cl, pH 8.5, then loaded on a Hybond N + nylon membrane (GE Healthcare, Little Chalfont, UK) using a 96-well dot-blot apparatus (Bio-Rad Laboratories, Hercules, CA). After hybridizing at 80°C for 30 min and blocking with 5% non-fat milk for 1 hr at room temperature, the membrane was incubated in polyclonal anti-5hmC antibody (Active Motif #39769, 1:10,000, Carlsbad, CA) at 4°C overnight. 5hmC was visualized by chemiluminescence. The densities of the dots on membrane were captured and quantified by ImageJ. To ensure equal loading, the membrane was stained with methylene blue post-immunoblotting. Statistical significance of differences in 5hmC content between treatments was assessed by Student $t$ test, at $\alpha = 0.05$.

## Immunofluorescence assay for 5hmc

Cells were seeded in 24-well plates with coverslips at 10,000–50,000 cells per well. After treatments, cells were fixed with 4% paraformaldehyde and then incubated with 2N HCl at 37°C for 20 min and neutralized with 100 µM Tris-HCl for 10 min. After washing with PBS and blocking with 0.4% Triton X-100 with 10% FBS in PBS for 1 hr, cells were incubated with anti-5-hmC antibody (Active Motif #39769, 1:1,000) at 4°C overnight. Alexa Fluor 488-conjugated donkey anti-rabbit IgG (1:500) was used as an immunofluorescent secondary antibody. Cells were then counterstained with DAPI.

## Intracellular labile fe(II) detection

Labile Fe(II) was detected using the reported methods (*Spangler et al., 2016*) and *Aron et al., 2016*). In the first method, cells were treated with puromycin conjugates TRX-Puro, Diox-Puro (a dioxolane conjugate as non-peroxidic control), or puromycin (1 µM) for 2 hr (*Spangler et al., 2016*). After washing with PBS, cells were fixed in 4% paraformaldehyde for 15 min at room temperature. Cells were then permeabilized with 0.4%Triton X-100 with 10% FBS and 0.25% fish skin gelatin in PBS for 30 min. Cells were incubated with anti-puromycin antibody (1:500) (Kerafast, Boston, MA) overnight at 4°C. After washing, cells were incubated with anti-mouse Alexa Fluor 488 secondary antibody (1:500) for 1 hr. Cells were then counterstained with DAPI.

Labile Fe(II) was also measured by FIP-1 probe as described (*Aron et al., 2016*). Briefly, cells were treated with or without 8-CPT-cAMP (10, 100 µM) for 4 hr. After wash, 10 µM FIP-1 in HBSS was added for 90 min incubation followed with HBSS (containing calcium and magnesium) wash. Cells were maintained in HBSS at 37°C during acquiring images using a Zeiss laser confocal microscope 710. FIP-1 was excited using a 488 nm laser ('Green' channel and 'FRET' channel). 'Green' emission was collected using a META detector between 500 and 535 nm, 'FRET' emission was collected using a META detector between 555 and 611 nm. Differential interference contrast (DIC) Image was also collected. Analysis and quantification was performed using ImageJ. Statistical analyses for multiple comparisons were carried out through one-way ANOVA with the Bonferroni correction using the software R.

## Endocytotic vesicle pH detection

The acidification of endocytotic vesicles was measured by pHrodo Red Dextran (Thermo Scientific, Waltham, MA) according to the manufacturer's instructions. Briefly, after plating and cAMP treatments, cells were washed three times with PBS and incubated with pHrodo Red Dextran at a final concentration of 20 µg/ml for 20 min in the incubator. Cells were then washed again with PBS and fixed with 4% PFA for 10 min. After fixation, cells were counterstained with Hoechst 33342. Images were captured by a Zeiss LSM 710 confocal microscopy.

## In vitro PKA assay

To confirm the activity of PKA inhibitors, a non-radioactive PepTag PKA assay (Promega, Madison, WI) was used to measure PKA activity. A2058 cells were plated at 10 cm dishes and and Schwann cell plated in 24 well plates, which were incubated with or without PKA inhibitors (H-89 and KT5720) for 30 min. After treatment with cAMP (100 µM) for 30 min at 37°C, cells were harvested for lysates and for intracellular labile (FeII) detection. The lysate was incubated with the PepTag peptide (Leu-Arg-Arg-Ala-Ser-Leu-Gly), which is a substrate of PKA for phosphorylation. The phosphorylated peptides have a negative charge that was then separated from the non-phosphorylated peptide by agarose gel electrophoresis. The bands were excised and the phosphorylated peptide was quantified by reading the absorbance at 570 nm using a Synergy 2 Biotek spectrophotometer.

## Gene silencing by siRNA

Small short interfering RNA (siRNA) was used to transiently silence Ferritin heavy chain 1 (FTH1) and RAPGEF2. To decrease the expression of Ferritin, a pool of three siRNA duplexes (cat# SR301663, Origene Technologies, Rockville, MD) against FTH1 and scramble siRNA (cat# SR30004) were used. To reduce the expression of RapGEF2, a pool of four siRNA duplexes (cat# L-009742-00-0005, Dharmacon, Lafayette, CO) was applied. siRNA were delivered to cells by SiTran Transfection reagent (Origene Technologies). After transfection for 72 hr, cells were treated with cAMP for intracellular labile Fe(II) measurement. Cells in a subset of wells were harvested for RNA and protein extraction and subsequent qRT-PCR and immunoblot assays.

## Immunoblot

Cells were washed twice with PBS and then lysed with RIPA buffer (50 mM Tris-HCl, 150 mM NaCl, 0.1% SDS, 0.5% sodium deoxycholate, 1% NP40) in the presence of protease and phosphatase inhibitors. Prior to SDS-PAGE, cell lysates were re-suspended in SDS sample buffer (60 mM Tris–HCl, 1% SDS, 10% glycerol, 0.05% bromophenol blue, pH 6.8, with 2% β-mercaptoethanol). Samples were subjected to 10% SDS-PAGE (Bio-Rad, Hercules, CA) and transferred to PVDF membranes (Bio-Rad) for immunoblot. Transfer efficiency was determined by Ponceau S staining (Sigma-Aldrich, St. Louis, MO). PVDF membranes were incubated with blocking solution (TBS containing 0.1% Tween 20% and 5% BSA) and were probed with specific antibodies including anti-Ferritin heavy chain (B-12) (sc-376594, Santa Cruz Biotechnology, Dallas, TX) and anti-β-Actin (13E5) (cat# 4970, Cell Signaling Technology, Danvers, MA). Protein bands were detected using a chemiluminescence kit (Millipore, Billerica, MA).

## Image acquisition and analysis

Cell fluorescence images were acquired into a 512 × 512 frame size by averaging 16 times at a bit depth of 8 using a Zeiss LSM 710 confocal microscope (Oberkochen, Germany). Fluorescence intensity was quantified using Fiji (ImageJ) (*Schindelin et al., 2012*). Average intensity values were measured from every cell within the image field from a minimum of five 20 × images per condition (Around 450 cells per condition). The intensity values from individual cells were plotted and statistically analyzed by one-way *ANOVA* with Tukey post hoc test using GraphPad Prism seven from GraphPad Software (La Jolla, CA).

## Quantitative real-time RT-PCR

RNA was extracted from cultured Schwann cells using RNeasy kits from Qiagen (Hilden, Germany). A nanodrop 8000 photospectrometer (Thermo Scientific, Waltham, MA) was used to quantify RNA. The qScript Flex cDNA kit from Quanta Biosciences (Beverly, MA) was used for reverse transcription (RT) according to the manufacturer's instructions. Quantitative real-time RT-PCR (qRT-PCR) was performed in triplicate on a QuantStudio 12K Flex using the PowerUp Sybr Green Master Mix from Life Technologies (Carlsbad, CA). Primers were designed to span introns (*Figure 1—source data 1*). The transcript amplification results were analyzed with the QuantStudio 12K Flex software, and all values were normalized to the levels of *Sdha* using the $2^{-(\Delta\Delta Ct)}$ method. Statistical significance of differences in expression levels was assessed by Student *t* test, at $\alpha=0.05$.

## RNA-seq

Total RNA was extracted from cells using the RNEasy Mini Kit from Qiagen (Hilden, Germany). A Bioanalyzer 2000 was used to measure the quality of RNA. All samples' RNA integrity numbers (RIN) were above 9. Whole transcriptome sequencing (also known as RNA-seq) was carried out at the Sequencing Core of John P. Hussman Institute of Human Genomics at the University of Miami using the TruSeq Stranded Total RNA Library Prep Kit from Illumina (San Diego, CA). RNA-seq was performed with three replicates per group to give ~89% power to detect a 1.5-fold change between and a standard deviation of 14.5% of the mean. This standard deviation was calculated from actual data used for analysis. Briefly, after ribosomal RNA (rRNA) was depleted, sequencing libraries were ligated with standard Illumina adaptors and subsequently sequenced on a Hiseq2000 sequencing system (125 bp paired-end reads, four samples per lane; Illumina, San Diego, CA, USA). All samples after sequencing had between 39,094,152 and 54,392,564 reads. Raw read data was first run through quality control metrics using FastQC (http: //www.bioinformatics.babraham.ac.uk/projects/fastqc/). Reads were trimmed with trim_galore (http://www.bioinformatics.babraham.ac.uk/projects/trim_galore/) to remove low-quality bases from reads (scores < 20 in Phred + 33 format) and Illumina adapters. After quality control was checked and trimming performed, sequence reads were aligned to the rat transcriptome (Rnor_6.0, Ensembl.org) and quantified using the STAR aligner (*Dobin et al., 2013*), and normalized using CuffQuant and CuffNorm, which are part of the Tuxedo Suite (*Trapnell et al., 2012*). All samples had between 37,714,718 and 52,162,956 aligned reads. Statistical significance was determined using three alternative differential expression calculators: edgeR, DESeq2, and CuffDiff (*Robinson et al., 2010*; *Love et al., 2014*; *Trapnell et al., 2012*). To reduce false positives, differentially expressed features were determined by cutoff adjusted *P* values below 0.05 across all three methods (false discovery rate, FDR). DESeq2 called 8589 differential transcripts, edgeR called 8029 differential transcripts, and CuffDiff called 8076 differential transcripts with a total of 7020 transcripts called as differential by all three programs. Of the 7020 transcripts, 3809 transcripts increase in expression and 3211 transcripts decrease after treatment with cAMP. Differential transcripts were visualized using z-scores per transcript where blue represents lower expression and red represents higher expression in the form of a heatmap generated using the heatmap.2 function of gplots (https://cran.r-project.org/web/packages/gplots/index.html).

## hMeDIP-seq

Genomic DNA was extracted from Schwann cells using QIAamp DNA mini kits from Qiagen (Hilden, Germany) according to the manufacturer's instructions. A Qubit Fluorometer from Life Technologies (Carlsbad, CA, USA) was used to quantify the concentration of DNA. A Bioanalyzer 2000 was used to measure the quality of DNA. DNA was submitted for hydroxymethylated DNA

immunoprecipitation sequencing (hMeDIP-seq) at the Epigenomics Core at the University of Michigan (*Mohn et al., 2009*). hMeDIP-seq was performed with three replicates per group to give ~98% power to detect a 2-fold change between groups and a standard deviation of 19% of the mean. This standard deviation was calculated from actual data used for analysis. Briefly, DNA is sonicated to approximately 100 bp and then ligated with Illumina adaptors. A portion of the DNA was set aside for sequencing as unprecipitated input DNA and the remaining DNA was incubated overnight at 4 ˚C with an antibody against 5hmC from Active Motif (Carlsbad, CA). The antibodies, along with immunoprecipitated DNA, were then pulled out of solution using Protein G magnetic beads from Invitrogen, (Carlsbad, CA). Magnetic beads were then washed with immunoprecipitation buffer (10 mM sodium phosphate pH 7.0 with 140 mM NaCl and 0.05% triton x-100). The beads were then resuspended in proteinase K buffer and incubated for 3 hr at 55 ˚C to remove the antibodies from the DNA. Unprecipitated input DNA was incubated with proteinase K buffer alongside the precipitated DNA. After proteinase K treatment, DNA was purified using AMPure beads from Beckman Coulter (Brea, California). Efficiency of immunoprecipitation was evaluated using the 5hmC, 5mC, cytosine DNA standard pack from Diagenode (Searing, Belgium). DNA was then sequenced on a HiSeq4000 sequencing system (50 bp single-end reads, three samples per lane; Illumina, San Diego, CA). Immunoprecipitated samples all had between 123,192,512 and 143,992,231 reads and unprecipitated input samples all had between 84,030,758 and 142,427,775 reads.

Reads were trimmed with trim_galore to remove low quality bases from reads (scores < 20 in Phred + 33 format), and Illumina adapters. After quality control was checked sequence reads were aligned to the rat genome (Rnor_6.0, Ensembl.org) using BWA (*Li and Durbin, 2009*). All immunoprecipitated samples had between 117,762,176 and 137,740,430 aligned reads and unprecipitated input samples had between 80,822,191 and 138,629,799 aligned reads. Multimapped reads were removed using Samtools (*Li et al., 2009b*) and duplicate reads were removed using PicardTools (https://broadinstitute.github.io/picard/). All precipitated samples had between 63,471,316 and 78,618,595 uniquely aligned reads and all unprecipitated input samples had between 44,195,649 and 80,394,782. To identify regions of the genome with substantial levels of 5hmC, peaks were called and filtered using the Irreproducible Discovery Rate (IDR) method developed for the ChIP-seq portion of the ENCODE project (*Li et al., 2011*). Briefly, peak calling was performed with MACS2 using the narrow peak mode and a relaxed threshold of 0.001 (*Zhang et al., 2008*). The number of peaks called for the control samples were 364,776, 346,424, and 331,841. The number of peaks called for the cAMP samples were 367,389, 356,223, and 351,249. Peaks were ranked from strongest to weakest peaks in each individual sample, and the rank order of common peaks among samples was compared to filter for only peaks with similar strength across all samples using the R-scripts provided for the IDR pipeline with a threshold of 0.02. After filtering for peaks in common, 106,807 peaks remained for the cAMP samples and 120,057 peaks remained for the control samples. The lower number of filtered peaks among the cAMP samples is likely a result of treatment causing higher variability in 5hmC levels in samples. When peaks across sample types were merged, a total of 192,143 peaks remained to be investigated in all samples. Reads within peak regions were quantified using HT-Seq-count (*Anders et al., 2015*). Statistical significance was determined using edgeR as was used for RNA-seq, except without using the calcNormFactors function so as to normalize to total read counts rather than counts within peaks. This normalization was necessary to account for a global change in 5hmC levels. Differential expression was also calculated using DESeq2. To minimize false positives, we considered only peaks with a minimum of 2X fold change and below an adjusted *P*-value (false discovery rate, FDR) below 0.05 by both edgeR and DESeq2. By this method, 66,963 peaks were upregulated, and 10,026 peaks were downregulated after treatment with cAMP, and 115,154 peaks remained unchanged.

## Sequencing reads visualization

For each individual sample, coverage at each base in the genome was calculated and normalized to total read count using bedtools genomecov (http://bedtools.readthedocs.io/en/latest/content/tools/genomecov.html). Coverage files were converted to bigwig files using the wigToBigWig program from UCSC genome browser (http://hgdownload.cse.ucsc.edu/admin/exe/linux.x86_64.v287/). Read density for peaks in each sample was visualized using version 2 of DeepTools (*Ramírez et al., 2016*) and individual bigwig files. Visualization of read density across differential genes was also performed using DeepTools and the merged bigwig files.

## Integration of hMeDIP-seq and RNA-seq

5hmC peaks were assigned to regions of the genome using Region_analysis in the diffReps package (*Shen et al., 2013*). Peaks whose center was within 3,000 bp of the transcription start site (TSS) of a given gene were considered promoter regions for that gene, and peaks whose center lay in the remainder of the gene or up to 1,000 bp following the transcription end site (TES) were considered gene body for that gene. Peaks outside of gene body regions and gene promoter regions were classified as intergenic. Differential transcripts, with differential peaks assigned to either the promoter region or gene body of that gene, were considered to be potentially regulated by 5hmC. Of the 7020 differential transcripts, 4071 (58%) were found to be potentially regulated by 5hmC. Further, visualization of both RNA-seq and hMeDIP-seq data was done using UCSC genome browser (https://genome.ucsc.edu/) and bigwig files showing loci of differential transcripts.

It is known that elevation of cAMP regulates the activity of three different transcription factors: CREB1, CREM and ATF1. The Ma'ayan laboratory has posted analyzed data online from both the ChEA and ENCODE projects (http://amp.pharm.mssm.edu/X2K/#downloads). To find genes potentially regulated by these three transcription factors, we used the ENCODE 2015 and ChEA 2015 databases (*Yip et al., 2012*; *Lachmann et al., 2010*). Experiments using rat cells are uncommon in these databases, so we combined the lists of genes found regulated by all experiments of a given transcription factor from human, mouse, or rat experiments. ATF1 was found to regulate 2000 genes across a variety of human cells in the ENCODE projects, of which 1996 genes were found to have a rat homolog in the Jackson Labs Complete Homology Class report (http://www.informatics.jax.org/homology.shtml). CREM was found to regulate 5776 genes in mouse testicular cells (GC1-SPG) in the ChEA project, of which 5773 were found to have a rat homolog in the Jackson Labs Complete Homology Class report. CREB1 was found to regulate 4040 genes across human leukemia (K562) and adenocarcinoma (A549) cells in the ENCODE project, as well as 957 genes in human embryonic kidney (HEK-293) cells in the ChEA project. CREB1 was also found to regulate 2393 genes in rat hippocampus and 3057 genes in mouse testicular cells (GC1-SPG) in the ChEA project. Combining data from all CREB1 experiments in both ChEA and ENCODE gave 6997 rat genes potentially regulated by CREB1. Combining results from all experiments of all three transcription factors, 9767 genes have been found to potentially be regulated by at least one of these three transcription factors. Of the 7020 differential transcripts, 3965 (56.5%) were included in the list of genes potentially regulated by one of the three transcription factors that is activated by cAMP/PKA. Differences in transcription can be possibly attributed to both transcription factors and changes in 5hmC for 2372 (33.8%) of the 7020 transcripts and 1356 (19.3%) are not likely regulated by either transcription factor or 5hmC.

## Acknowledgements

This work is supported by NIH grants (R01NS089525, R21CA191668 to GW and GM079465 to CJC) and a Craig H Neilsen Foundation grant (M1501061 to PVM). We thank Mr. Chris Gustafson for his technical support on dot blot. GW is also supported by the Dr. Nasser Ibrahim Al-Rashid Orbital Vision Research Center at University of Miami. PVM is supported by The Miami Project to Cure Paralysis and The Buoniconti Fund. CJC is an Investigator with the Howard Hughes Medical Institute and a CIFAR Senior Fellow. ATA. thanks the NSF for a graduate fellowship and was partially supported by a Chemical Biology Training Grant from the NIH (T32 GM066698).

## Additional information

### Funding

| Funder | Grant reference number | Author |
| --- | --- | --- |
| National Institute of Neurological Disorders and Stroke | R21CA191668 | Gaofeng Wang |
| National Cancer Institute | R01NS089525 | Gaofeng Wang |
| National Institute of General Medical Sciences | GM079465 | Christopher J Chang |

The funders had no role in study design, data collection and interpretation, or the decision to submit the work for publication.

## Author contributions

Vladimir Camarena, Validation, Investigation, Visualization, Methodology, Writing—review and editing; David W Sant, Formal analysis, Validation, Investigation, Visualization, Writing—review and editing; Tyler C Huff, Formal analysis, Visualization, Writing—review and editing; Sushmita Mustafi, Validation, Investigation, Writing—review and editing; Ryan K Muir, Resources, Writing—review and editing; Allegra T Aron, Validation, Writing—review and editing; Christopher J Chang, Adam R Renslo, Paula V Monje, Resources, Data curation, Writing—review and editing; Gaofeng Wang, Conceptualization, Resources, Data curation, Supervision, Funding acquisition, Investigation, Methodology, Writing—original draft, Project administration, Writing—review and editing

## Author ORCIDs

Vladimir Camarena (iD) https://orcid.org/0000-0001-9466-6863
Gaofeng Wang (iD) http://orcid.org/0000-0001-8202-8282

## Decision letter and Author response

Decision letter https://doi.org/10.7554/eLife.29750.031
Author response https://doi.org/10.7554/eLife.29750.032

# Additional files

**Supplementary files**
• Transparent reporting form
DOI: https://doi.org/10.7554/eLife.29750.028

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
