## [Decision Letter]

Thank you for submitting your article "cAMP Signaling Regulates DNA Demethylation by Augmenting the Intracellular Labile Ferrous Iron Pool" for consideration by *eLife*. Your article has been reviewed by three peer reviewers, and the evaluation has been overseen by a Reviewing Editor and Kevin Struhl as the Senior Editor. One of the reviewers, Yehia Daaka, has agreed to reveal his identity.

The reviewers have discussed the reviews with one another and the Reviewing Editor has drafted this decision to help you prepare a revised submission.

Summary:

In this paper, Camarena et al. examine the contribution of extracellular cues to the regulation of epigenetic modifications, focusing on GPCR-based regulation of the Tet DNA demethylase. The authors delineate a mechanism by which GPCR signaling via cAMP induces endosome acidification, leading to the release of iron, which serves as a cofactor for Tet enzymes. This pathway operates independently of classical cAMP effectors, but requires RapGEF2 for endosome acidification. This work highlights an unappreciated role for GPCR signaling in regulating DNA methylation and gene expression.

The reviewers generally agree that this work is likely to be of significant interest to the community. Because of the potential of this work to be paradigm-shifting in the cAMP and DNA methylation fields, it is important that the authors further solidify their conclusions. The reviewers raised some major concerns which should be successfully addressed before the manuscript can be considered for publication.

Essential revisions:

1) The authors have not adequately entertained alternative hypotheses, nor tested all assumptions regarding their own model. Specifically, the reviewers suggested that GPCR signaling might affect nuclear import of iron, and/ or processing of 5hmC to 5-formylcytosine or 5-carboxycytosine. Furthermore, the authors have not demonstrated that free iron (and endosome acidification) are limiting for Tet activity. The alternative hypotheses should be ruled out, and the idea that endosome acidification and iron availability are limiting demonstrated (reviewer #1 has suggestions on addressing these issues).

2) There are serious concerns from multiple reviewers regarding the dosage and duration of different treatments. Particularly concerning were the experiments done after 7 days of cAMP treatment, as physiological GPCR signaling is rapid and transient, and cells are likely to lose viability after 7 days of treatment. Questions regarding the type of cAMP analog used and its concentration, as well as the effects of the various activators/ inhibitors on cAMP, Fe++, and 5hmC levels at different timepoints, should be fully addressed.

3) Along the same lines, termination of the cAMP response has not been addressed. It would be important to show downregulation of the response by conducting a timecourse of Fe++ and 5hmC levels after forskolin washout. A similar timecourse experiment in the presence of cAMP should be done to show any sensitization effects, which are hinted at in Figure 2.

4) Additional mechanistic insight into RapGEF2 function should be provided. The authors should address whether the endosome acidification and LIP are mediated by Rap, and if so, which isoform. Secondly, the authors should address how RapGEF2 is affected by cAMP, e.g. by checking expression levels and subcellular localization.

5) The sequencing analysis should be done more thoroughly. The data should be collected at 48-72 hrs after cAMP treatment, and compared to the data collected at 7 days. Sequencing should consider methylation of promoters, enhancers, and gene bodies separately.

6) To increase confidence in the results, the authors should use complementary approaches (ideally something other than microscopy-based assays) whenever possible. Reviewer #3 has given suggestions on how to accomplish this.

*Reviewer #1:*

This is an interesting manuscript that describes enhanced DNA 5hmC modification induced by cAMP. The authors showed that cAMP promotes iron uptake through enhanced acidification of endosomes. The increase cellular labile iron(II) promotes TET activity on 5mC oxidation. There are two major weaknesses need to be addressed:

1) "In healthy cells such as primary cultured Schwann 125 cells, the most plausible rate-limiting factor for Tet's enzymatic activity is the replenishment of the Fe(II) cofactor rather than 2OG, a relatively abundant intermediate of the Krebs cycle."

I agree 2OG may not be limiting factor but iron(II) is quite abundant as well. The abundance of nuclear iron(II) has not been adequately studied as iron(II) needs to be transported into nucleus in order to supply TET. This is maybe something the authors should really look into with the cAMP signaling. The authors should really isolate the nuclear fraction and use ICP-MS to study total iron level instead of whole cell. Or at least perform imaging of labile iron within nucleus. The technique I recommend is the synchrotron based X-ray fluorescence metal imaging.

The acidification of endosome is a good mechanism but I doubt it is the only one or the one that matters. Under different cellular signaling iron enriches in nucleus. cAMP might influence that pathway, which explains less effect observed with ferritin KD. If simply for the level of cytoplasmic iron level ferritin should have a major role.

2) The studies of 5hmC with gene expression changes are shallow. The authors mixed different effects here. Gene body 5hmC levels should correlate with mRNA level changes. Promoter 5hmC levels could go either way. Another place to really check is enhancer 5hmC level changes. More in depth analysis would help.

*Reviewer #2:*

The manuscript by Camarena and colleagues examines contribution of extracellular cues to epigenetic modifications and the mechanisms involved using the specific case of Gs-coupled GPCR-regulated activation of 5hmC. Using genetic, biochemical, and cellular approaches, the authors delineate a signaling pathway showing increased intracellular cAMP levels lead to increased LIP through endosome acidification and RapGEF2. Based on experimental results, the authors conclude that augmented cAMP levels (including by Gs-coupled receptors) increase LIP and DNA demethylation. Although conclusions of this interesting report are supported by experimental results, few issues need to be addressed:

What is the effect of cAMP accumulation on 5hmC expression at early time points (say 1-24 hrs)?

Chemical reagents concentrations used (cAMP and inhibitors) are noticeably high (up to 100 microM); why? Also, in Figure 1 the cAMP concentrations needed for (presumably maximal effects on) MEFs are lower than for other cell types. Any explanation?

The authors suggest that RapGEF2 regulates LIP in a mechanism that is dependent upon cAMP. What is the evidence for this conclusion? Also, does RapGEF2 promote activation of Rap? Other small GTPases? If Rap is the effector of RapGEF2, it would be interesting to identify what isoform of Rap is involved in the cAMP-induced LIP, and whether this effect is through endosome acidification?

*Reviewer #3:*

In their study Camarena et al. investigate the role of cAMP in DNA methylation. Using mainly imaging approaches the authors conclude that cAMP enhances the generation of 5-hydroxymethylcytosine (5hmC) (an indicator of DNA methylation/demethylation status) which results in significant changes in the programs of gene transcription of Schwann cells. The authors provide evidence suggesting that the increase in 5hmC levels in response to cAMP elevations may be a generalised event (Schwann; HEK; MEF and SYSH5 cells). The effects of cAMP on 5hmC levels seen independent of the classical cAMP effectors (PKA; EPACs etc.) and the ten-eleven translocation (Tet) proteins. On the contrary, the authors propose that cAMP enhances endosome acidification via RapGEF2 resulting in an increase in the levels of Fe(II) and essential cofactor of the Tet enzymes.

This is an interesting and thought provoking article that would be of interest in the readership of *eLife*. However, at its present state the article presents a major issue: the timing and intensity of the cAMP (or cAMP triggering) treatments used, are not well defined and are far from being physiological, undermining the core hypothesis of the existence of a GPCR-cAMP-RapGEF2 axis.

1) cAMP treatments:

First of all the authors use the cell "permeant" cAMP analog 8-CPT-cAMP in a range from 1µM to 250 µM. Since it is not specified the type of 8-CPT-cAMP used (Na^+^ salt Vs AM-ester) it is very difficult to understand the levels of the messenger at the cellular level. For instance if they used Na^+^ salt in concentrations of low µM the cellular levels of 8-CPT cAMP would be too low to activate any effector, on the contrary at 1µM of the AM-ester the levels of cellular 8-CPT-cAMP would reach saturating values making difficult to reconcile the dose response data observed by the authors (Figure 1, Figure 2).

Secondly in addition to 8-CPT-cAMP the authors use other means to increase cellular cAMP levels. The concentration of FSK (100µM) is far too much (saturating doses go from 10 to 25µM) and the effect observed is basically the same as the treatments with IBMX (100µM) to block PDEs, caffeine to block a subset of PDEs (and release Ca^2+^ from the stores) and Bicarbonate (50mM) to activate only the soluble adenylyl cyclase that would produce a fraction (if any) of the FSK increases in cellular cAMP. The authors should measure the cellular levels of cAMP reached with these treatments (elisa or FRET-based assays).

2) Timing:cAMP signals are physiologically produced by GPCRs and are meant to be fast and transient. On the contrary here the authors treat their cells with 100µM 8-CPT-cAMP from 1 to 7 days (Figure 1). The max effect on 5hmC levels is reached after 24h (Figure 1) and is maintained for 7 days. These data are not well matched to the data of Figure 2 showing that the levels of Fe(II) peaked after 4h trending to decrease after that (what is the level of Fe(II) after 7days?). This is an important point since the authors for their RNA-seq experiments choose to treat their cells for 7 days with 100µM 8CPT-cAMP.

3) Termination/feedback

In order to be conceived as a signalling cascade the GPCR-cAMP-RapGEF2 axis should undergo termination as well. This is something that is not suggested by the experiments on Figure 1 where the levels of 5hmC remain high for the whole period of the treatment. On the contrary the trent of the Fe(II) levels in Figure 2 would suggest a negative feedback. The authors should perform a series of experiments addressing this point. For instance they could treat cells for 2-4 hours with FSK and then follow the levels of 5hmC and Fe(II) for a period of time (up to 24-48 hours) would be suggested. This is also relevant for the data presented in Figure 6 where treatment with extremely high doses of isoproterenol induces Fe(II) and 5hmC increases after 2 days. This is somewhat surprising since β receptors would most likely be internalised and desensitised with prolong treatments with saturating (1-2µM) doses of Isoproterenol. This is a crucial point, as it is not conceivable that once activated this pathway cannot be turned off.

[Editors' note: further revisions were requested prior to acceptance, as described below.]

Thank you for resubmitting your work entitled "cAMP Signaling Regulates DNA Demethylation by Augmenting the Intracellular Labile Ferrous Iron Pool" for further consideration at *eLife*. Your revised article has been favorably evaluated by Kevin Struhl (Senior Editor) and a Reviewing Editor.

The manuscript has been improved but there are some remaining issues that need to be addressed before acceptance, as outlined below:

1) The authors have shown intriguing new results that termination of GPCR signaling does not restore 5hmC levels to their basal state. The authors should solidify this finding by repeating the forskolin washout experiment in HEK293 cells, and assaying 5hmC levels at 72 hours. Based on the outcome of this experiment, the authors should provide a commentary as to the role of GPCR signaling in establishment vs. maintenance of 5hmC in different cell types.

2) In light of the above finding, the title should be made more accurate by referring to "5hmC" rather than "demethylation," as the authors have not demonstrated full removal of the methyl mark in this system. Something like "cAMP signaling regulates DNA 5hmC by augmenting the intracellular labile ferrous iron pool" would more accurately reflect the results.

3) The authors write that the timepoints for sequencing analysis were chosen to allow analysis of the effects of GPCR signaling on myelinating genes. However, they mention only 2 individual genes involved in this process. The authors should use the sequencing data to provide a global analysis of myelinating genes, and any others involved in Schwann cell differentiation.

---

## [Author Response]

Essential revisions:

1) The authors have not adequately entertained alternative hypotheses, nor tested all assumptions regarding their own model. Specifically, the reviewers suggested that GPCR signaling might affect nuclear import of iron, and/ or processing of 5hmC to 5-formylcytosine or 5-carboxycytosine. Furthermore, the authors have not demonstrated that free iron (and endosome acidification) are limiting for Tet activity. The alternative hypotheses should be ruled out, and the idea that endosome acidification and iron availability are limiting demonstrated (reviewer #1 has suggestions on addressing these issues).

We thank the reviewers for raising this concern. The hydroxylation of 5mC to 5hmC requires the following: the enzyme Tet methylcytosine dioxygenases (expressed in the cell), the substrate 5mC (abundant in the genome), the co-substrate 2OG, the cofactor Fe(II), and oxygen. If the limiting factor is not Fe(II), the alternative hypothesis could be that this chemical reaction is limited by 2OG and oxygen. However, under cell culture conditions, 2OG, which is a relatively abundant intermediate in the Krebs cycle, and oxygen are available to Tet. Despite their abundance, 5hmC was barely detectable in untreated cells (Figure 1), suggesting that the halted hydroxylation catalyzed by Tet is not limited by 2OG or oxygen.

We and others have shown that ascorbate promotes the conversion of 5mC to 5hmC (Minor EM et al., JBC2013; Yin R et al., Blaschke K et al., Nature 2013; J Am Chem Soc 2013; Chen J et al., Nat Genet. 2013; Dickson KM et al., BBRC 2013). While 5hmC was nearly undetectable in untreated cells, ascorbate induced a strong increase of 5hmC in Schwann cells (Figure 1). Ascorbate has the capacity of reducing the catalytically inactive Fe(III)/Fe(IV) to Fe(II), which underlies the 5hmC increase, just as it does for collagen hydroxylases. These results suggest that Fe(II) is likely the limiting factor for 5hmC generation.

To further test whether Fe(II) is the limiting factor for 5hmC generation, we treated Schwann cells with Fe(II). Schwann cells cultured in media containing Fe(III) showed low basal levels of 5hmC. Addition of Fe(II) to the media induced 5hmC generation in the cells (see Author response image 1). These results suggest that Fe(II) is likely the limiting factor for 5hmC generation.

Robust 5hmC signal could be detected after cells were treated with cAMP (Figure 1). The affinity of Tet for Fe(II) (KD) is about 0.41 ± 0.05 μM (Hore TA et al., PNAS 2016). TRX-puro probe detects labile Fe(II) in 0.1 ~ 1 μM range (Spangler et al.,Nat Chem Biol. 2016), which corresponds to intracellular Fe(II) levels (Epsztejn S., et al. Blood1999). Using TRX-puro probe, we have detected obvious increases of labile Fe(II) by cAMP treatment, suggesting that an increase in labile Fe(II) over basal levels could mediate the effect of cAMP on 5hmC by activating Tet activity.

Furthermore, this effect of cAMP on 5hmC was largely abolished by treatment with two different iron chelators (Figure 3), suggesting that the promotion of 5hmC by cAMP is dependent on iron. Endosome acidification is one key step for iron uptake. We showed that cAMP increased endosome acidification and that inhibition of V-ATPase diminished the effect of cAMP on labile Fe(II) and 5hmC (Figure 4). Taken together, these data suggest that labile Fe(II) is one key limiting factor for Tet activity.

While total iron could be abundant in the cells, labile Fe(II) must be tightly controlled due to its active redox activity, especially its ability to produce free radicals through the Fenton reaction (Dunn LL, et al., Trends Cell Biol. 2007). Currently, measuring labile Fe(II) in the cell still remains an experimental challenge. Even with these difficulties, we managed to show that cAMP elevated the total labile Fe(II) using Trx-Puro probes in the cell. To further confirm these results, we now show an obvious increase of labile Fe(II) in HEK-293 cells after cAMP treatment using another newly developed labile Fe(II) probe FIP-1 (Aron AT, et al., J Am ChemSoc.2016) in collaboration with Christopher Chang (UC Berkeley/HHMI). This new data is now included in Figure 2—figure supplement 3.

Little is known how labile Fe(II) enters the nucleus. It is generally accepted that there is a rapid equilibrium between the cytoplasmic labile pool and the nuclear labile Fe(II) pool, presumably via nuclear pores (Ma Y et al., Metallomics 2015), suggesting that cAMP could potentially elevate labile Fe(II) in the nucleus. Following the reviewer’s suggestion, we assessed total iron in the fractionated nuclear and cytoplasm extracts from HEK-293 cells treated with or without cAMP by ICP-MS. This experiment was completed by Christopher Chang’s team. The results indicate that there is no significant change in total cellular iron after treatment with cAMP. It is noteworthy that the labile iron pool is only a very small part (0.2 ~0.3%) of total iron in the cell (Petrat F et al., Biol Chem. 2002). Therefore, it is reasonable that cAMP elevates labile Fe(II) but might not exert a detectable change in the total iron.

Labile Fe(II) is a cofactor for Tet methylcytosine dioxygenases, which can further oxidize 5hmC to 5fC and 5caC. Due to the fact that 5fC/5caC are rare in the genome and are relatively difficult to be accurately evaluated, we focused on 5hmC in the study. Because the principle of these Tet-mediated oxidation reactions is similar, it is plausible that cAMP promotes the conversion of 5mC to 5hmC, and further to 5fC and 5caC as well, which will be examined in our future studies.

2) There are serious concerns from multiple reviewers regarding the dosage and duration of different treatments. Particularly concerning were the experiments done after 7 days of cAMP treatment, as physiological GPCR signaling is rapid and transient, and cells are likely to lose viability after 7 days of treatment. Questions regarding the type of cAMP analog used and its concentration, as well as the effects of the various activators/ inhibitors on cAMP, Fe++, and 5hmC levels at different timepoints, should be fully addressed.

Activation of Gs-coupled receptors elevates intracellular cAMP principally from these three steps: (1) activation of membrane bound AC; (2) Gβγ subunit activation of certain AC isoforms; (3) Continuous production of cAMP by Gs and AC in the endosome after receptor internalization. Thus, intracellular cAMP elevation by activated Gs-coupled receptors persists by compartmentalized signaling. Furthermore, under physiological conditions, the stimuli are often persistent and periodic, which can result in a long-term oscillation of intracellular cAMP (Dyachok O, et al., Nature 2006).

It has been shown that the differentiation of Schwann cells to a myelinating phenotype can be induced only by prolonged treatment with cAMP. After differentiation, cells remain viable as shown by specific marker expression (Monje PV et al., Glia. 2009). This correlates with the in vivo evidence that GPR126, via elevation of cAMP, promotes myelination (Monk KR, et al., Science2009). Furthermore, in our initial experiments, Schwann cells had been treated with cAMP for up to 15 days and remained healthy as observed by Incucyte video and images.

The cAMP analog used was 8-CPT-cAMP (sodium salt), which is cell permeable and mimics the action of cAMP in the cell as do other cAMP analogues such as 8-CPT-cAMP-AM. We thank the reviewer’s insightful comments about membrane permeability of 8-CPT-cAMP (sodium salt). It is now clear to us that the permeability of 8-CPT-cAMP is only about 22% of what is applied extracellularly (Wener K et al., Naunyn Schmiedebergs Arch Pharmacol. 2011). Even with this less-than-ideal permeability, 1 μM 8-CPT-cAMP still caused an obvious elevation of labile Fe(II). By examining the dose-dependent effect of 8-CPT-cAMP on labile Fe(II) and 5hmC, we showed that 8-CPT-cAMP at 100 μM induced robust increases in both labile Fe(II) and 5hmC. Since 8-CPT-cAMP at 100 μM promotes cultured Schwann cells into a promyelinating phenotype (Monje PV et al., Glia. 2009), we then selected this treatment for high-throughput sequencing.

Forskolin treatment was initially carried out at 100 μM because forskolin at 100 μM produces a maximal increase of intracellular cAMP in Schwann cells (Yoshimura T et al., Neurochemical Research 1994). Following the reviewer’s suggestion, we have replicated the effect of forskolin at much lower concentrations (5 and 10 μM) on 5hmC (now included in Figure 1—figure supplement 2) and on labile Fe(II) (now included in Figure 3—figure supplement 1). Bicarbonate was used at 50 μM which was shown to induce a maximal effect on cAMP (Chen Y et al., Science 2000). Caffeine and IBMX concentrations were chosen at 100 μM to inhibit phosphodiesterases based on previous publications (Fredholm BB et al., Pharmacological Reviews 1999). The effect of cAMP was further verified by stimulation of Gs-coupled receptors.

We also include new data showing that short term treatment (such as 1 h) and lower concentrations of cAMP (10 μM) and forskolin (5 or 10 μM) elevated 5hmC in the cell (now included in Figure 1—figure supplement 2). These data suggest that the effect of cAMP on Fe(II) is likely rapid and is responsible for activating Tet to produce 5hmC. However, our ability to observe changes in the LIP at very early time points (< 2 h) is limited with TRX-puro. This is because the rate-limiting step in puromycin release is a β-elimination reaction that occurs subsequent to reaction of TRX-puro with Fe(II) (see Spangler et al., Nat Chem Biol. 2016). We believe that the delay in peak Fe(II) signal compared to 5hmC signal is significantly a result of puromycin release kinetics rather than a delayed mobilization of Fe(II).

3) Along the same lines, termination of the cAMP response has not been addressed. It would be important to show downregulation of the response by conducting a timecourse of Fe++ and 5hmC levels after forskolin washout. A similar timecourse experiment in the presence of cAMP should be done to show any sensitization effects, which are hinted at in Figure 2.

Following the reviewers’ suggestion, we conducted the following experiments.

a) HEK-293 cells were treated with cAMP (10 μM) for 1, 4, or 24 h and cells were washed immediately after the completion of treatments. 24 h following washout, 5hmC increases were detectable in all three treatments (now included in Figure 1—figure supplement 2).

b) Schwann cells were treated with forskolin (10 μM) for 3 h followed by washout. The results showed that compared to non-treated cells, labile Fe(II) was elevated after treatment for 3 h and then declined to basal levels (now included in Figure 3—figure supplement 1). In contrast, the elevation of 5hmC sustained for 24 h after washing in Schwann cells (now included in Figure 1—figure supplement 2).

c) Schwann cells were treated with forskolin (10 μM) for 4, 12, 24 h followed by washout and 5hmC measurement at 72 h following treatment. We found that 5hmC levels with these treatments after washout were comparable to 72 h of continuous treatment (now included in Figure 1—figure supplement 2).

Results of these experiments suggest that labile Fe(II) elevation is relatively transient but the increase in 5hmC persists after termination of forskolin or cAMP by washout.

4) Additional mechanistic insight into RapGEF2 function should be provided. The authors should address whether the endosome acidification and LIP are mediated by Rap, and if so, which isoform. Secondly, the authors should address how RapGEF2 is affected by cAMP, e.g. by checking expression levels and subcellular localization.

We performed additional experiments to look into the mechanistic role of RapGEF2. Rap is the major downstream effector of RapGEF2. We knocked down the expression of Rap isoforms (Rap1 and Rap2). The results showed that silencing Rap1, but not Rap2, largely abolished the effect of cAMP on endosome acidification and labile Fe(II), suggesting that Rap1 is likely one major effector mediating the effect of RapGEF2 on the LIP (Figure 5—figure supplement 3).

cAMP has been shown to activate RapGEF2, which in turn activates Rap1 (Emery AC, et al. Sci Signal. 2013). Rap1 presents in the plasma membrane and endosomes. Further investigations are needed to study the detailed role of RAP1 in endosome acidification.

5) The sequencing analysis should be done more thoroughly. The data should be collected at 48-72 hrs after cAMP treatment, and compared to the data collected at 7 days. Sequencing should consider methylation of promoters, enhancers, and gene bodies separately.

The differentiation of Schwann cells into a promyelinating phenotype requires a prolonged treatment with cAMP (Monje PV. Glia 2009). We thus selected cells treated for 7 days for RNA-seq in order to show the impact of cAMP on the transcriptome, but more importantly, on the expression of myelinating genes.

Following the reviewers’ helpful suggestion, we further examined 5hmC-enriched regions in the promoters and gene bodies (this new data is now included in Figure 8—figure supplement 1). The results showed that 5hmC increased in the gene bodies globally but the increase was much more dramatic in transcriptionally upregulated genes compared to downregulated or nondifferential genes. There was no obvious change of 5hmC in the promoters of transcriptionally unchanged genes. However, 5hmC increased in the promoters of both upregulated and downregulated genes. Due to the lack of relevant ChIP-seq data (such as H3K4me1, H3K4me3, H3K27ac, and H3K27me3) in Schwann cells, we are unable to determine enhancer regions for analysis.

6) To increase confidence in the results, the authors should use complementary approaches (ideally something other than microscopy-based assays) whenever possible. Reviewer #3 has given suggestions on how to accomplish this.

We agree with the reviewers. We used another probe FIP-1 developed in collaborator Christopher Chang lab (UC Berkeley/HHMI) (Aron AT, et al., *J* Am Chem Soc.2016). Distinct from Trx-Puro probes, this method measures FRET signaling from the probe specifically disrupted by ferrous labile iron. The experiments were performed entirely in the Chang lab and results showed an obvious increase of labile Fe(II) in HEK-293 cells after cAMP treatment using labile Fe(II) (now included in Figure 2—figure supplement 3). Other than these methods, we have not established other sensitive and reliable techniques to accurately quantify the LIP in the cell.

Reviewer #1:

This is an interesting manuscript that describes enhanced DNA 5hmC modification induced by cAMP. The authors showed that cAMP promotes iron uptake through enhanced acidification of endosomes. The increase cellular labile iron(II) promotes TET activity on 5mC oxidation. There are two major weaknesses need to be addressed:1) […] The acidification of endosome is a good mechanism but I doubt it is the only one or the one that matters. Under different cellular signaling iron enriches in nucleus. cAMP might influence that pathway, which explains less effect observed with ferritin KD. If simply for the level of cytoplasmic iron level ferritin should have a major role.

Please see our response to Essential revisions #1.

2) The studies of 5hmC with gene expression changes are shallow. The authors mixed different effects here. Gene body 5hmC levels should correlate with mRNA level changes. Promoter 5hmC levels could go either way. Another place to really check is enhancer 5hmC level changes. More in depth analysis would help.

We thank the reviewer for the insightful comments. We further examined 5hmC-enriched regions in the promoters and gene bodies (this new data is now included in Figure 8—figure supplement 1). The results showed that 5hmC increased in the gene bodies globally but the increase was much more dramatic in transcriptionally upregulated genes compared to downregulated or nondifferential genes. There was no obvious change of 5hmC in the promoters of transcriptionally unchanged genes. However, 5hmC increased in the promoters of both upregulated and downregulated genes. Due to the lack of relevant ChIP-seq data (such as H3K4me1, H3K4me3, H3K27ac, and H3K27me3) in Schwann cells, we are unable to determine enhancer regions for analysis.

Reviewer #2:

The manuscript by Camarena and colleagues examines contribution of extracellular cues to epigenetic modifications and the mechanisms involved using the specific case of Gs-coupled GPCR-regulated activation of 5hmC. Using genetic, biochemical, and cellular approaches, the authors delineate a signaling pathway showing increased intracellular cAMP levels lead to increased LIP through endosome acidification and RapGEF2. Based on experimental results, the authors conclude that augmented cAMP levels (including by Gs-coupled receptors) increase LIP and DNA demethylation. Although conclusions of this interesting report are supported by experimental results, few issues need to be addressed:What is the effect of cAMP accumulation on 5hmC expression at early time points (say 1-24 hrs)?

Following the reviewer’s suggestion, we now include new data showing that forskolin (10 μM) treatment for 3 h elevates 5hmC level (now included in Figure 1—figure supplement 2). Furthermore, 5hmC is also increased 24 h after a brief (1 h) cAMP (10 μM) treatment (now included in Figure 1—figure supplement 2). These data suggest that the effect of cAMP could be fast but still needs time for labile Fe(II) elevation, which further activates Tet to generate 5hmC.

Chemical reagents concentrations used (cAMP and inhibitors) are noticeably high (up to 100 microM); why? Also, in Figure 1 the cAMP concentrations needed for (presumably maximal effects on) MEFs are lower than for other cell types. Any explanation?

It is now clear to us that the permeability of 8-CPT-cAMP is only about 22% of what is applied extracellularly (Wener K et al., Naunyn Schmiedebergs Arch Pharmacol. 2011). Even with this less-than-ideal permeability, 1 μM 8-CPT-cAMP still caused an obvious elevation of labile Fe(II). By examining the dose-dependent effect of 8-CPT-cAMP on labile Fe(II) and 5hmC, we showed that 8-CPT-cAMP at 100 μM induced robust increases in both labile Fe(II) and 5hmC. Since 8-CPT-cAMP at 100 μM promotes cultured Schwann cells into a promyelinating phenotype (Monje PV, et al., Glia.2009), we then selected this treatment for high-throughput sequencing.

Forskolin treatment was initially carried out at 100 μM because forskolin at 100 μM produces a maximal increase of intracellular cAMP in Schwann cells (Yoshimura T et al., Neurochemical Research 1994). Following the reviewer’s suggestion, we have replicated the effect of forskolin at much lower concentrations (5 and 10 μM) on 5hmC (now included in Figure 1—figure supplement 2) and on labile Fe(II) (now included in Figure 3—figure supplement 1). Bicarbonate was used at 50 μM which was shown to induce a maximal effect on cAMP (Chen Y et al., Science 2000). Caffeine and IBMX concentrations were chosen at 100 μM to inhibit phosphodiesterases based on previous publications (Fredholm BB et al., Pharmacological Reviews 1999). The effect of cAMP was further verified by stimulation of Gs-coupled receptors.

We also include new data showing that short term treatment (such as 1 h) and lower concentrations of cAMP (10 μM) and forskolin (5 or 10 μM) elevated 5hmC in the cell (now included in Figure 1—figure supplement 2). These data suggest that the effect of cAMP on Fe(II) is likely rapid and is responsible for activating Tet to produce 5hmC. We treated different cell types with cAMP at the same concentration (10 μM) to show that the effect of cAMP on 5hmC is not limited to Schwann cells. The difference in 5hmC response to cAMP treatment, to our understanding, could be due to the variations in iron uptake (such as transferrin receptor density), Tet expression levels, and DNA methylation (5mC) levels among these cells.

The authors suggest that RapGEF2 regulates LIP in a mechanism that is dependent upon cAMP. What is the evidence for this conclusion? Also, does RapGEF2 promote activation of Rap? Other small GTPases? If Rap is the effector of RapGEF2, it would be interesting to identify what isoform of Rap is involved in the cAMP-induced LIP, and whether this effect is through endosome acidification?

cAMP has been shown to activate RapGEF2, which in turn activates Rap1 (Emery AC, et al., Sci Signal. 2013). We showed that knockdown of RapGEF2 largely abolishes the upregulation of the LIP by cAMP (Figure 5). In contrast, inhibition of the canonical PKA pathway does not interfere with the elevation of the LIP by cAMP (Figure 5). These results suggest that RapGEF2 likely mediates the effect of cAMP on the LIP.

We performed additional experiments to look into the mechanistic role of RapGEF2. Rap is the major downstream effector of RapGEF2. We therefore knocked down the expression of Rap isoforms (Rap1 and Rap2). The results showed that silencing Rap1 largely abolished the effect of cAMP on endosome acidification and labile Fe(II), suggesting that Rap1 is likely one major effector mediating the effect of RapGEF2 on the LIP (Figure 5—figure supplement 3).

Reviewer #3:

*[…] This is an interesting and thought provoking article that would be of interest in the readership of* eLife. However, at its present state the article presents a major issue: the timing and intensity of the cAMP (or cAMP triggering) treatments used, are not well defined and are far from being physiological, undermining the core hypothesis of the existence of a GPCR-cAMP-RapGEF2 axis.1) cAMP treatments:First of all the authors use the cell "permeant" cAMP analog 8-CPT-cAMP in a range from 1µM to 250 µM. Since it is not specified the type of 8-CPT-cAMP used (Na^+^ salt Vs AM-ester) it is very difficult to understand the levels of the messenger at the cellular level. For instance if they used Na^+^ salt in concentrations of low µM the cellular levels of 8-CPT cAMP would be too low to activate any effector, on the contrary at 1µM of the AM-ester the levels of cellular 8-CPT-cAMP would reach saturating values making difficult to reconcile the dose response data observed by the authors (Figure 1, Figure 2).Secondly in addition to 8-CPT-cAMP the authors use other means to increase cellular cAMP levels. The concentration of FSK (100µM) is far too much (saturating doses go from 10 to 25µM) and the effect observed is basically the same as the treatments with IBMX (100µM) to block PDEs, caffeine to block a subset of PDEs (and release Ca^2+^ from the stores) and Bicarbonate (50mM) to activate only the soluble adenylyl cyclase that would produce a fraction (if any) of the FSK increases in cellular cAMP. The authors should measure the cellular levels of cAMP reached with these treatments (elisa or FRET-based assays).

We thank the reviewer insightful comments about the membrane permeability of 8-CPT-cAMP. 8-CPT-cAMP is a cell-permeable sodium salt which mimics the action of cAMP in the cell as do other cAMP analogues such as 8-CPT-cAMP-AM. It is now clear to us that the permeability of 8-CPT-cAMP is only about 22% of what is applied extracellularly (Wener K et al., Naunyn Schmiedebergs ArchPharmacol. 2011). Even with this less-than-ideal permeability, 1 μM 8-CPT-cAMP still caused an obvious elevation of labile Fe(II). By examining the dose-dependent effect of 8-CPT-cAMP on labile Fe(II) and 5hmC, we showed that 8-CPT-cAMP at 100 μM induced robust increases in both labile Fe(II) and 5hmC. Since 8-CPT-cAMP at 100 μM promotes cultured Schwann cells into a promyelinating phenotype (Monje PV, et al., Glia. 2009), we then selected this treatment for high-throughput sequencing.

Forskolin treatment was initially carried out at 100 μM because forskolin at 100 μM produces a maximal increase of intracellular cAMP in Schwann cells (Yoshimura T et al., Neurochemical Research 1994). Following the reviewer’s suggestion, we have replicated the effect of forskolin at much lower concentrations (5 and 10 μM) on 5hmC (now included in Figure 1—figure supplement 2) and on labile Fe(II) (now included in Figure 3—figure supplement 1). Bicarbonate was used at 50 μM which was shown to induce a maximal effect on cAMP (Chen Y et al., Science 2000). Caffeine and IBMX concentrations were chosen at 100 μM to inhibit phosphodiesterases based on previous publications (Fredholm BB et al., Pharmacological Reviews 1999). The effect of cAMP was further verified by stimulation of Gs-coupled receptors.

2) Timing:cAMP signals are physiologically produced by GPCRs and are meant to be fast and transient. On the contrary here the authors treat their cells with 100µM 8-CPT-cAMP from 1 to 7 days (Figure 1). The max effect on 5hmC levels is reached after 24h (Figure 1) and is maintained for 7 days. These data are not well matched to the data of Figure 2 showing that the levels of Fe(II) peaked after 4h trending to decrease after that (what is the level of Fe(II) after 7days?). This is an important point since the authors for their RNA-seq experiments choose to treat their cells for 7 days with 100µM 8CPT-cAMP.

Activation of Gs-coupled receptors elevates intracellular cAMP principally from these three steps: (1) activation of membrane bound AC; (2) Gβγ subunit activation of certain AC isoforms; (3) Continuous production of cAMP by Gs and AC in the endosome after receptor internalization. Thus, intracellular cAMP elevation by activated Gs-coupled receptors persists by compartmentalized signaling. Furthermore, under physiological conditions, the stimuli are often persistent and periodic, which can result in a long-term oscillation of intracellular cAMP (Dyachok O, et al., Nature 2006).

It has been shown that the differentiation of Schwann cells to a myelinating phenotype can be induced only by prolonged treatment with cAMP. After differentiation, cells remain viable as shown by specific marker expression (Monje PV, et al., Glia. 2009). This correlates with the in vivo evidence that GPR126, via elevation of cAMP, promotes myelination (Monk KR, et al., Science2009). Furthermore, in our initial experiments, Schwann cells had been treated with cAMP for up to 15 days and remained largely healthy as observed by Incucyte video and images.

Following the reviewer’s suggestion, we now include new data showing that forskolin (10 μM) treatment for 3 h elevates 5hmC level (now included in Figure 1—figure supplement 2). Furthermore, 5hmC is also increased 24 h after a brief (1 h) cAMP (10 μM) treatment (now included in Figure 1—figure supplement 2). These data suggest that the effect of cAMP could be fast but still needs time for labile Fe(II) elevation, which further activates Tet to generate 5hmC. To understand the impact of 5hmC elevation on transcription, we treated Schwann cells for 7 days based on previous publications that myelinating genes are only expressed after a prolonged treatment (up to 10 days) with cAMP.

3) Termination/feedbackIn order to be conceived as a signalling cascade the GPCR-cAMP-RapGEF2 axis should undergo termination as well. This is something that is not suggested by the experiments on Figure 1 where the levels of 5hmC remain high for the whole period of the treatment. On the contrary the trent of the Fe(II) levels in Figure 2 would suggest a negative feedback. The authors should perform a series of experiments addressing this point. For instance they could treat cells for 2-4 hours with FSK and then follow the levels of 5hmC and Fe(II) for a period of time (up to 24-48 hours) would be suggested. This is also relevant for the data presented in Figure 6 where treatment with extremely high doses of isoproterenol induces Fe(II) and 5hmC increases after 2 days. This is somewhat surprising since β receptors would most likely be internalised and desensitised with prolong treatments with saturating (1-2µM) doses of Isoproterenol. This is a crucial point, as it is not conceivable that once activated this pathway cannot be turned off.

Please see our response to Essential revisions #3.

[Editors' note: further revisions were requested prior to acceptance, as described below.]

The manuscript has been improved but there are some remaining issues that need to be addressed before acceptance, as outlined below:1) The authors have shown intriguing new results that termination of GPCR signaling does not restore 5hmC levels to their basal state. The authors should solidify this finding by repeating the forskolin washout experiment in HEK293 cells, and assaying 5hmC levels at 72 hours. Based on the outcome of this experiment, the authors should provide a commentary as to the role of GPCR signaling in establishment vs. maintenance of 5hmC in different cell types.

We have replicated the forskolin washout experiment in HEK-293 cells. The results showed that forskolin treatment (24 hours) followed by washout induced 5hmC elevation at levels comparable to continuous treatment (72 hours) in HEK-293 cells. In contrast, 5hmC levels largely retreated to the base line in shorter forskolin treatments (4, 12 hours) followed by washout. A parallel experiment showed that forskolin treatment (4 hours) followed by washout induced 5hmC elevation at levels comparable to continuous treatment (24 hours) in HEK-293 cells. Compared to Schwann cells, we observed that after treatment of HEK-293 cells followed by washout and assessed at 72 hours, 5hmC levels retreated toward the base line. It is known that 5hmC is not maintained in DNA synthesis and HEK-293 cells replicate much faster than the slowly-dividing primary Schwann cells. Taken together, these results suggest that high levels of 5hmC in cells could be established by cAMP signaling. However, the long-term maintenance of 5hmC requires continuous cAMP signaling. These data are now included in Figure 1—figure supplement 3.

2) In light of the above finding, the title should be made more accurate by referring to "5hmC" rather than "demethylation," as the authors have not demonstrated full removal of the methyl mark in this system. Something like "cAMP signaling regulates DNA 5hmC by augmenting the intracellular labile ferrous iron pool" would more accurately reflect the results.

The title is now changed to “cAMP Signaling Regulates DNA hydroxymethylation by Augmenting the Intracellular Labile Ferrous Iron Pool”.

3) The authors write that the timepoints for sequencing analysis were chosen to allow analysis of the effects of GPCR signaling on myelinating genes. However, they mention only 2 individual genes involved in this process. The authors should use the sequencing data to provide a global analysis of myelinating genes, and any others involved in Schwann cell differentiation.

We included a global analysis of GPCR signaling on myelinating genes or genes involved in Schwann cell differentiation. Our analysis showed that the transcription of 19 genes with known functions in regulating Schwann cell myelination was affected by cAMP treatment. Overall, 13 pro-myelinating genes were upregulated and 6 anti-myelinating genes were downregulated by cAMP treatment. Furthermore, we observed obvious 5hmC peak changes in 10 genes, suggesting the transcription of these genes could be correlated with cAMP-induced DNA hydroxymethylation. For the other 9 genes, their transcription could be regulated by PKA-targeted transcription factors. An alternative explanation could be due to the limitation of hMeDIP-seq by sequencing 5hmC enriched genomic regions using anti-5hmC antibody. These data are now included in Figure 8—source data 1.